# Tempora: Cell trajectory inference using time-series single-cell RNA sequencing data

**Thinh N. Tran**[1,2¤], **Gary D. Bader**[1,2,3]*

**1** Department of Molecular Genetics, University of Toronto, Ontario, Canada, **2** Donnelly Centre for Cellular and Biomolecular Research, University of Toronto, Ontario, Canada, **3** Department of Computer Science, University of Toronto, Ontario, Canada

¤ Current address: Gerstner Sloan Kettering Graduate School of Biomedical Sciences, New York, NY, United States of America

* gary.bader@utoronto.ca

## Abstract

Single-cell RNA sequencing (scRNA-seq) can map cell types, states and transitions during dynamic biological processes such as tissue development and regeneration. Many trajectory inference methods have been developed to order cells by their progression through a dynamic process. However, when time series data is available, most of these methods do not consider the available time information when ordering cells and are instead designed to work only on a single scRNA-seq data snapshot. We present Tempora, a novel cell trajectory inference method that orders cells using time information from time-series scRNA-seq data. In performance comparison tests, Tempora inferred known developmental lineages from three diverse tissue development time series data sets, beating state of the art methods in accuracy and speed. Tempora works at the level of cell clusters (types) and uses biological pathway information to help identify cell type relationships. This approach increases gene expression signal from single cells, processing speed, and interpretability of the inferred trajectory. Our results demonstrate the utility of a combination of time and pathway information to supervise trajectory inference for scRNA-seq based analysis.

## Author summary

Single-cell RNA sequencing (scRNA-seq) enables an unparalleled ability to map the heterogeneity of dynamic multicellular processes, such as tissue development, tumor growth, wound response and repair, and inflammation. Multiple methods have been developed to order cells along a pseudotime axis that represents a trajectory through such processes using the concept that cells that are closely related in a lineage will have similar transcriptomes. However, time series experiments provide another useful information source to order cells, from earlier to later time point. By introducing a novel use of biological pathway prior information, our Tempora algorithm improves the accuracy and speed of cell trajectory inference from time-series scRNA-seq data as measured by reconstructing known developmental trajectories from three diverse data sets. By analyzing scRNA-seq data at the cluster (cell type) level instead of at the single-cell level and by using known pathway information, Tempora amplifies gene expression signals from one cell using

**Data Availability Statement:** All source codes and validation data used in this manuscript are available at https://github.com/BaderLab/Tempora.

**Funding:** This work was supported by NRNB (U.S. National Institutes of Health, National Center for

Research Resources grant number P41 GM103504) to G.D.B. The funders had no role in study design, data collection and analysis, decision to publish, or preparation of the manuscript.

**Competing interests:** The authors have declared that no competing interests exist.

similar cells in a cluster and similar genes within a pathway. This approach also reduces computational time and resources needed to analyze large data sets because it works with a relatively small number of clusters instead of a potentially large number of cells. Finally, it eases interpretation, via operating on a relatively small number of clusters which usually represent known cell types, as well as by identifying time-dependent pathways. Tempora is useful for finding novel insights in dynamic processes.

This is a *PLOS Computational Biology* Methods paper.

## Introduction

Dynamic tissue-level processes, such as development, aging and regeneration, are critical for multicellular organisms. Single-cell RNA sequencing (scRNA-seq) enables us to map the range of cell types and states in these processes at cellular resolution [1]. A single scRNA-seq snapshot can be used to infer lineage relationships between cell types and states [2]. Snapshot scRNA-seq studies have been used to investigate multiple aspects of development, including the early embryo, blood, different areas of the brain and more [3]. Even though snapshot scRNA-seq can provide novel insights into development, it has recognized limits [4], including that cell populations that appear earlier or later than the sampling time cannot be studied. Time-series scRNA-seq can address some of these limits and has been increasingly applied to study tissue development, including in cerebral cortex [5], kidney [6], and heart [7].

When using scRNA-seq to study dynamic processes, whether through snapshot or time-series experiments, it is of interest to order cells at different stages along an axis that represents how far along they are on the process under study, based on their transcriptional signatures. The ordering problem, commonly termed pseudotime ordering if it is inferred from data without a known temporal ordering, consists of two main parts: the identification of a trajectory representing the paths that cells follow, and the determination of pseudotime values for individual cells along this trajectory. This inferred trajectory enables us to study the sequential changes of gene expression during a process, as well as identify branches and instrumental genes at the branching points. More than 70 computational methods to order cells along pseudotemporal axes, known as trajectory inference methods, have been published, which employ different strategies to infer lineage and order cells [8]. Most trajectory inference methods are developed based on the basic premise that cells closer in developmental lineage have more similar gene expression signatures, thus a likely trajectory is a path through gene expression space that maximizes cell-to-cell similarity. Common strategies for trajectory construction include fitting a minimum spanning tree (MST), which connects all data points in a path that minimizes distance between points, or nonlinear dimensionality reduction that identifies a low-dimensional manifold that cells lie on. Monocle 1, a pioneering trajectory inference method, constructs a MST connecting all cells in a reduced dimension space, then determines the longest path through this tree as the backbone and orders cells along this path [9]. Some methods, including TSCAN and Slingshot, build the MST on cell cluster centroids, representing cell types and states, instead of individual cells, and project cells on the MST to determine their pseudotime values [10,11]. Other methods, such as PAGA and StemID, use graph theory methods, such as graph partition, to construct trajectories [12,13]. Expanding on the graph partition idea, Monocle 3, the latest version of Monocle, infers the possible paths cells can take

through a dynamic process by learning a principal graph on the coarse-grained trajectory constructed by PAGA [14]. Many available trajectory inference methods have been evaluated and integrated in Dyno, a platform that enables users to conveniently apply selected methods to their data [8].

While many scRNA-seq trajectory inference methods exist, few have been designed to consider time-series information. Trajectory inference methods that explicitly incorporate temporal information include Waddington-OT, which models cells' movement through dynamic processes using the optimal transport framework, and CSHMM, which uses a continuous-state hidden Markov model to assign cells to developmental paths [15,16]. These methods demonstrate the utility of time information for trajectory inference. Here, we hypothesize that trajectory inference can be improved by the combined use of time series information in ordering cells and of pathway information to both reduce noise in gene expression data and increase interpretability of the dynamic process under study. To test this, we developed the Tempora method to infer cell lineage maps from time-series scRNA-seq data using biological pathway information. Tempora works at the cell cluster level to align cell types and states (clusters) across time points using redundancy reduced pathway enrichment vectors, then infers trajectory relationships between these cell types using the available temporal ordering information. Evaluating Tempora on three diverse time-series scRNA-seq data sets using gold standards showed that our method outperforms established trajectory inference methods.

## Result

### Method overview

The Tempora method infers cell type-based trajectories from time-series scRNA-seq data. Tempora focuses on identifying how cell types are related across the entire time-series data set, based on the established assumption that cells with similar gene expression profiles are closer in the underlying cell lineage. After identifying cell type transcriptome similarity relationships, Tempora orders these links based on the time-series data. Cell types identified primarily in earlier time points are ordered earlier in the trajectory than those identified primarily in later time points. To build a more robust trajectory, less influenced by small outlier cell populations and low per cell sensitivity of current scRNA-seq experimental methods, Tempora first clusters cells with similar transcriptional signatures and infers a trajectory that connects cell clusters rather than individual cells. These clusters represent putative cell types, such as progenitors, immune cells, cardiomyocytes, or stable cell states (e.g. cycling, proliferating or metabolizing [2]). Second, to improve robustness of the trajectory relationship identification step, clusters are compared to each other based on their redundancy-reduced biological pathway enrichment profiles instead of individual gene expression profiles. This also helps improve biological interpretability of the trajectory result, as trajectory-related pathway expression patterns can be automatically identified.

Tempora takes as input a preprocessed gene expression matrix from a time-series scRNA-seq experiment and cluster labels for all cells. Tempora then calculates the average gene expression profiles, or centroids, of all clusters before transforming the data from gene expression space to pathway enrichment space using single-sample gene set variation analysis (GSVA) [17] (Fig 1). To focus on high variance and non-redundant pathway information, Tempora applies PCA on the pathway enrichment analysis result and selects important PCs using a scree plot (the scree plot is output to help the user select the number of PCs to use). Pathways with high (>0.4, as recommended by [18]) loadings on those PCs are used to construct the lineage in the next step.

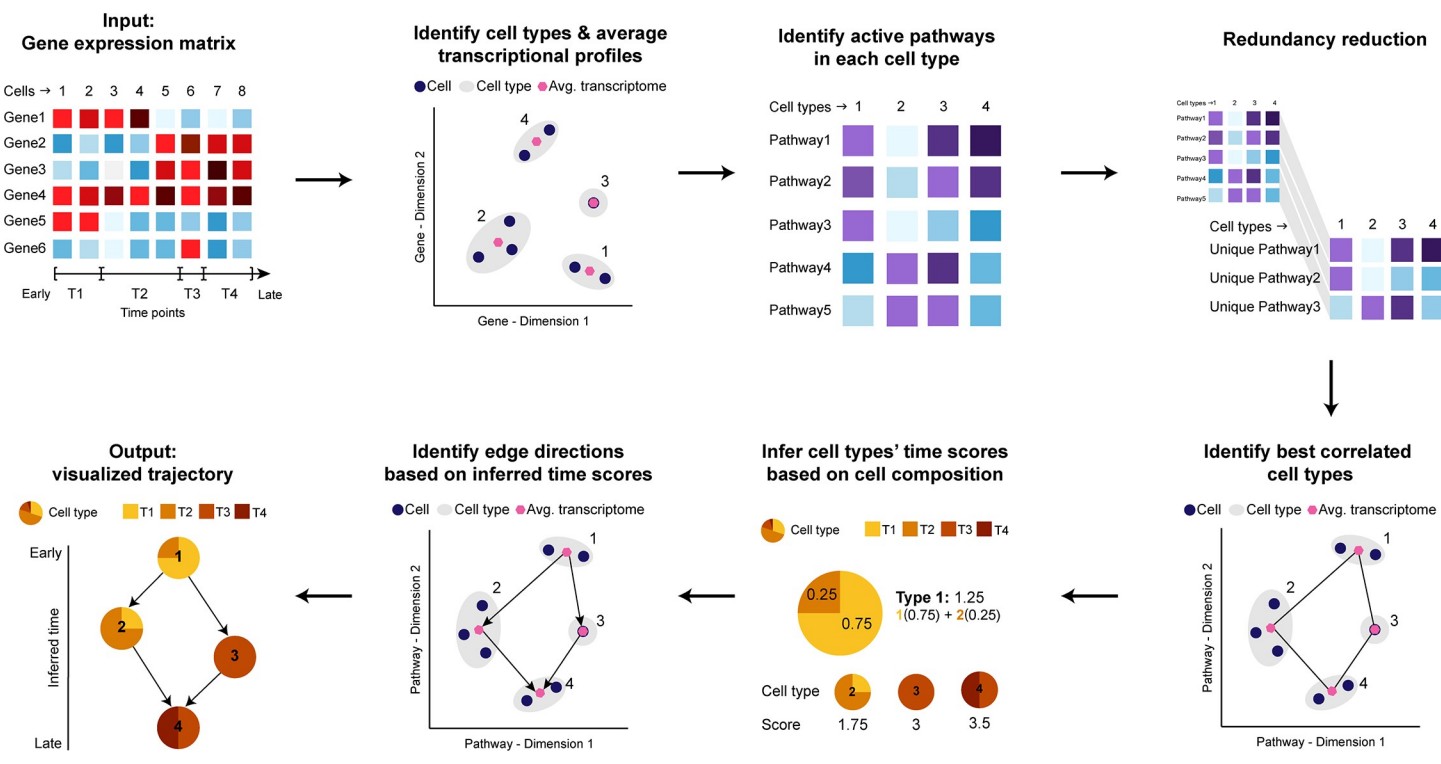

**Fig 1. Schematic of the Tempora algorithm.**

We abstract the trajectory as a network of cell clusters, where vertices represent the cell types or states identified as clusters and directed edges represent temporal transitions between types or states. To infer this network, Tempora uses the established ARACNE [19] method to identify cluster relationships based on mutual information (MI) of the cluster pathway enrichment profiles. ARACNE filters the MI network using the data processing inequality to remove edges with the smallest MI in all triples, which helps remove indirect connections (Fig 1). All resulting ARACNE edges are kept at this step. After constructing the trajectory, Tempora uses available temporal information from the input data to determine edge directions. First, each cluster is assigned a temporal score corresponding to its cell composition from each time point, so that a cluster containing more cells from an early time point will have a low score and vice versa. Trajectory network edges are then directed so that their sources have a lower temporal score than their targets, indicating a transition from an early cell type to a later cell type. The trajectory is visualized using the Sugiyama hierarchical layout algorithm [20].

Tempora includes a downstream pathway exploration tool to determine and visualize pathways that change significantly over the trajectory. These pathways are identified by fitting a generalized additive model to the enrichment scores of each pathway across all clusters and selecting pathways whose expression patterns deviate significantly from the null model of uniform pathway enrichment scores across all time points.

## Validation on human skeletal muscle myoblast time-series data

We evaluated Tempora's performance by inferring trajectories from a diverse set of time series scRNA-seq data sets and comparing them with known, gold standard trajectories that we manually curated from the literature (muscle and neural cortex data) or that accompanied the scRNA-seq data in original publications (brain cerebellum data). We first applied Tempora to

a small human skeletal muscle myoblast (HSMM) data set, which includes 271 cells collected at 0, 24, 48 and 72 hours after the switch of human myoblast culture from growth to differentiation media. At the optimal clustering resolution (see Methods), five clusters were identified and automatically annotated using GSVA, with known markers of proliferation (*CDK1*), muscle differentiation (*MYOG*) and a known population of contaminating myofibroblast cells (*SPHK1*) [9] (S1A–S1D Fig). Tempora identifies a branching trajectory connecting these clusters, rooted at the myoblast cluster that contains mostly cells at 0 hours after the media switch. This cluster leads to three separate branches, including a branch connected to the fibroblast cluster, one connected to the myotube cluster, and the last one connected to the partially differentiated myotube cluster via an intermediate cluster (Fig 2A). This branching trajectory agrees with the known biology of muscle differentiation *in vitro*, in which myoblasts proliferate and exit the cell cycle before differentiating into myotubes [9]. The fibroblast cluster contains equal proportions of cells from all time points and uniquely expresses myofibroblast markers (*SPHK1*). The equal numbers of cells from all time points in this cluster suggest that the contaminating cells were present in the earliest time point and persist in the culture over time, while its separation from the other two branches suggest that these cells do not participate in the differentiation process. Thus, Tempora identifies fibroblasts as a source of contamination in the myoblast culture, consistent with results from other trajectory inference methods [9,10] and from the literature [21]. Another branch in this trajectory connects the myoblast cluster to the myotube cluster, which contains *MYOG*-positive cells mostly at 48 and 72 hours. (Fig 2A). *MYOG* is a required transcription factor for the terminal differentiation of myoblasts into myotubes and is rapidly upregulated when myoblasts start to differentiate around day 2 *in vitro* [22]. Therefore, the appearance of *MYOG*-positive myotubes at 48 hours and their connection to the myoblasts cluster, as predicted by Tempora, aligns with previous findings in the literature. Finally, the myoblast cluster is also connected to an intermediate cluster, which contains 75% cells from two early time points, expresses lower levels of *CDK1* and does not express *MYOG* (Fig 2A). The low *CDK1* expression suggests that cells in this cluster have begun to exit the cell cycle to start differentiation, thus representing an intermediate state between proliferating myoblasts and differentiated muscles that is consistent with our understanding of muscle differentiation [23]. This intermediate cluster is predicted to give rise to a cluster of partially differentiated cells, which contains mostly cells from later time points and expresses low levels of the muscle-specific transcription factor *MYOG*. Since HSMM cultures have been noted to differentiate asynchronously and with less than 100% efficiency, cells in this partially

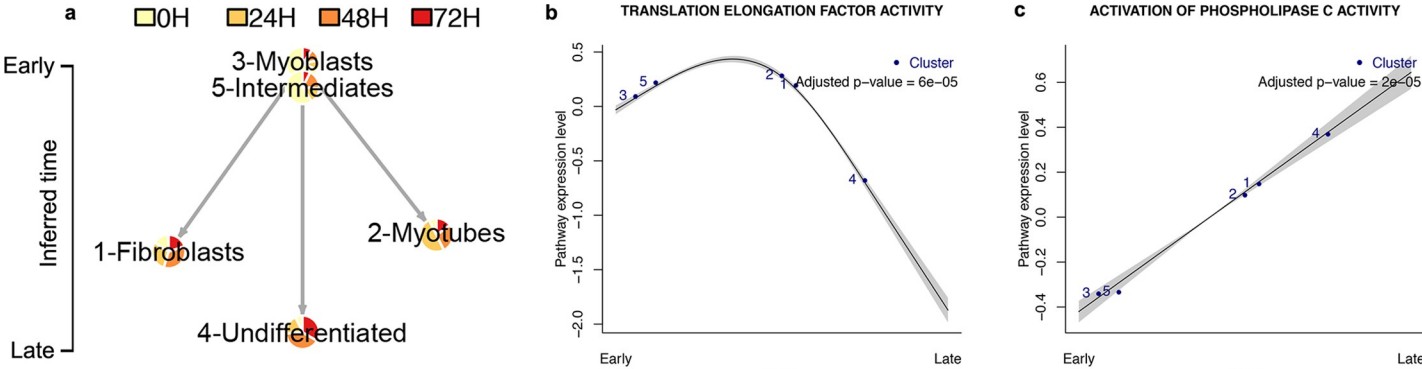

**Fig 2. Tempora analysis of the HSMM data set. a.** Tempora trajectory built on clusters in the HSMM data set. **b-c.** Time-dependent pathways in the HSMM data set identified by Tempora's pathway exploration feature. Blue dots represent clusters/cell types, the black line is the best fit of a generalized additive model (GAM) on the pathway enrichment scores of each pathway across all clusters, and grey areas depict 95% confidence intervals of the fitted model.

differentiated cluster likely represent a cell population that is slower to differentiate or failed to go through differentiation, as observed in previous studies [22]. Tempora, thus, predicts a branching trajectory that matches the structure and gene expression patterns of the known trajectory [23].

We used the pathway exploration feature of Tempora to identify pathways whose enrichment changed over time. Tempora identifies pathways that vary over time by fitting a generalized additive model (GAM) on the pathway enrichment scores of each pathway across all clusters/time and using ANOVA to compare the fitted model with the null model of uniform pathway enrichment over time. Pathways enriched early in the differentiation process include the cell cycle, biosynthesis and protein translation (Fig 2B and 2C). Pathways upregulated later are associated with the formation of myotubes, including morphogenesis and phospholipase C signaling, which regulate myogenic activity [23,24] (Fig 2B and 2C). Thus, Tempora's pathway exploration component can be used downstream of trajectory inference to identify pathways with interesting activity profiles over the time-series.

## Validation on murine cerebral cortex time-series data

We next applied Tempora on an embryonic murine cerebral cortex development scRNA-seq data set, which contains approximately 6,000 neural cells collected at embryonic days 11.5 (E11.5), E13.5, E15.5 and E17.5 [5] (Fig 3A). These cells cover a wide spectrum of neuronal development, from the early precursors (apical precursors (APs) and radial precursors (RPs)) to intermediate progenitors (IPs) and differentiated cortical neurons. Data at all time points were aggregated and batch effects were corrected with Harmony before clustering (see Methods) [25]. We annotated the seven resulting clusters automatically using GSVA and marker genes for APs, RPs, IPs, newborn neurons and neurons as done in the original publication (S1 Table). This resulted in the annotation of two AP/RP clusters mostly comprising cells at E11.5, which is consistent with the known emergence of RPs from APs at E11 [5,26], as well as two IP clusters, one IP/young neuron cluster and two neuron clusters, all of which contain cells from multiple time points as expected from their gradual specification over time [5] (S2B–S2G Fig).

Tempora predicts three trajectories, two rooted at the two AP/RP clusters and one rooted at an early IP cluster (Fig 3A). Each of the two AP/RP lineages has two branches: one terminating at an IP/young neuron cluster and another converging at a late neuron cluster. The lineage predicted by Tempora aligns with our understanding of AP/RP asymmetric division to generate IPs and neurons in early corticogenesis [5,26,27,28]. To better understand why there are two trajectories arising from two AP/RP clusters instead of one AP/RP cluster transforming into another AP/RP cluster in a single trajectory, we compared the gene expression profiles of the two clusters and identified cell cycle markers, such as *Mki67* and *Cdk1*, to be differentially expressed. This suggests that the two AP/RP clusters differ based on their cell cycle state: one is actively proliferating and expressing cell cycle markers while the other is not (S2D Fig), consistent with the known decreased proliferation of APs as they transition to RPs [5,29]. The observation that both AP/RP clusters contain equal proportion of cells from all time points suggest that these two proliferative and non-proliferative AP/RP populations arise before the time-series started, instead of one transforming into the other. Similarly, the IP cluster that serves as the root of the third trajectory contains many cells from the earliest time point and is thus unlikely to come from either of the AP/RP clusters, but may instead arise from earlier APs that are not captured in this time-series data. This IP cluster is predicted to give rise to a cluster of young neurons, which then mature into neurons as denoted by a dashed line in Fig 3A. The dashed line signifies very high temporal score similarity (>99%, default Tempora parameter) between the young neurons and neurons clusters indicating that we are not confident to assign

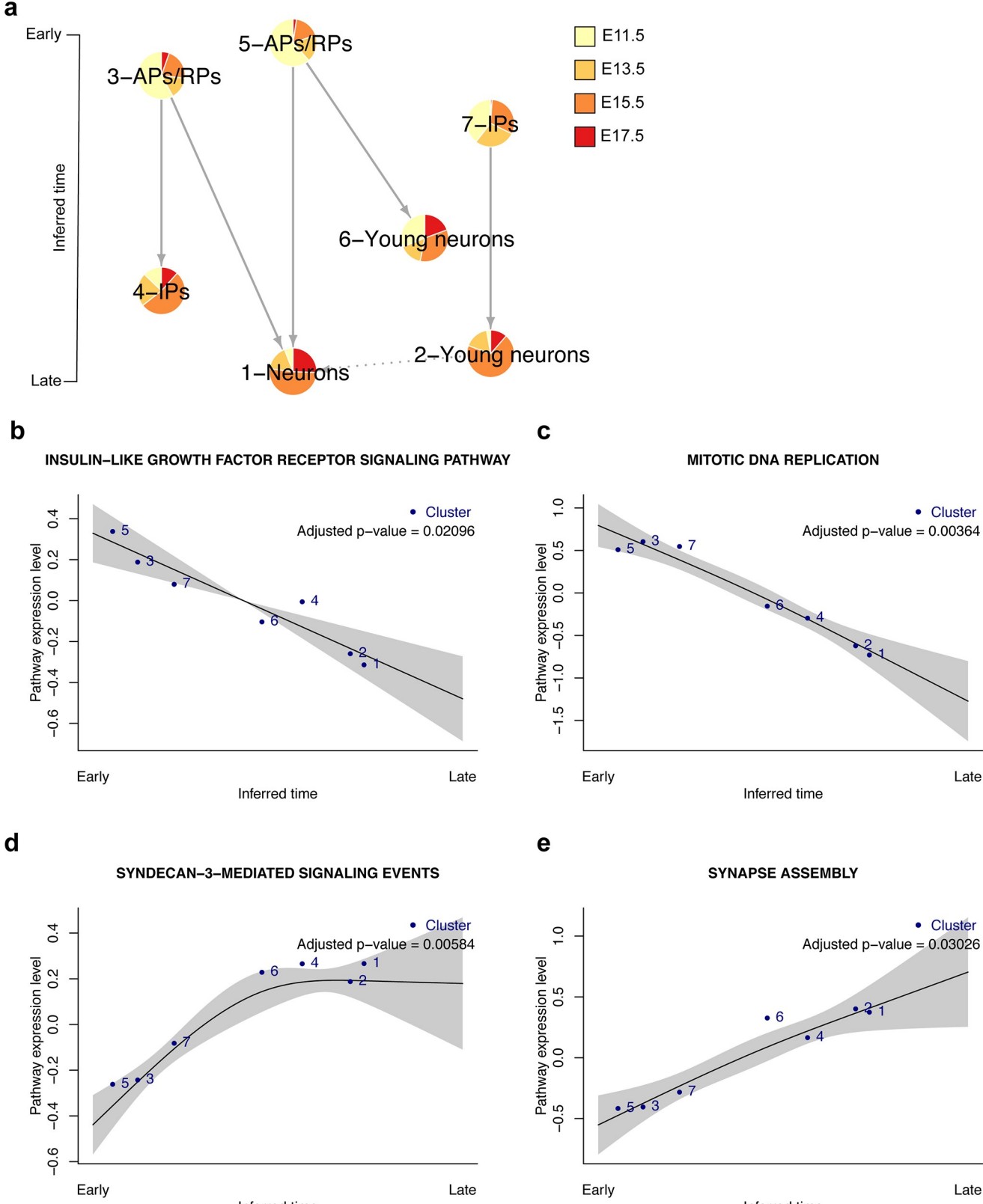

**Fig 3. Tempora analysis of the murine cortex data set. a.** Tempora trajectory built on clusters in the murine cerebral cortex data set. **b-e.** Time-dependent pathways in the murine cerebral cortex data set identified by Tempora's pathway exploration feature. Blue dots represent clusters/cell types, the

black line is the best fit of a generalized additive model (GAM) on the pathway enrichment scores of each pathway across all clusters, and grey areas depict 95% confidence intervals of the fitted model.

an edge direction based on time ordering. Overall, the transitions predicted by Tempora are consistent with our understanding of neurogenesis [5,26]. Tempora, thus, accurately identifies distinct trajectories originating from different populations in the murine cerebral cortex development data.

We used the pathway exploration feature of Tempora to analyze time-dependent pathway activity levels in the data. DNA replication and other mitotic pathways are enriched early (Fig 3B), while neuron-related pathways, such as synapse activity, dendritic morphogenesis and neurotransmitter synthesis, are enriched later (Fig 3D and 3E). These patterns are consistent with the known proliferation of neural progenitors at the beginning of neurogenesis [29] and neuronal activities of newborn neurons later in the process [30]. Tempora also identifies more subtle changes in signaling pathways over time, such as the early enrichment of Insulin-like growth factor signaling and the later upregulation of Syndecan-3 mediated signaling, both of which are consistent with their known roles in early neurogenesis and neural circuit assembly, respectively [31–33] (Fig 3B–3E).

## Validation on murine cerebellar development time-series data

Finally, to test Tempora's performance on a large data set containing cells from multiple lineages, we analyzed a murine cerebellar development time-series scRNA-seq data set of ~55,000 neural cells collected at nine time points (E10, E12, E14, E16, E18 and postnatal day 0 (P0), P5, P7 and P14) [34]. These cells were isolated from the mesial cerebellum of wild-type mice and belong to three main cell types found in the cerebellum: GABAergic neurons, glutamatergic neurons and glia. They span a large developmental spectrum from progenitors to fully differentiated neurons. From the original data set of ~60,000 cells, we removed ~5,000 non-neural cells, including mesenchymal stem cells, brainstem progenitors, endothelial cells, blood cells, meninges, pericytes and microglia to focus our analysis on the neural lineages only (see Methods). We aggregated the filtered data from all time points and batch-corrected with Harmony before clustering. Unsupervised clustering identified 24 clusters, many of which include cells from one or a few closely related time points (S3 Fig). Automated cluster labeling with GSVA and known marker genes (S1 Table) identified clusters of neural stem cells at the earliest time point, followed by the gradual emergence of cells belonging to the glutamatergic lineage, including embryonic granule cell progenitors (GCPs), upper rhombic lip cells (URLs) and nuclear transitory zone (NTZ) neurons, as well as ventricular zone (VZ) progenitors that give rise to GABAergic cells. In the late embryonic time points (E16-18), NTZ neurons and excitatory cerebellar nuclei neurons (ECNNs) in the glutamatergic lineage, as well as GABA interneurons and Purkinje cells in the GABAergic lineage start emerging. The glial lineage is the last to emerge, with gliogenic progenitors appearing during late embryonic time points and astrocytes appearing throughout postnatal time points. The annotations and timing of these clusters are as described in the original publication [34].

Tempora predicts three main trajectories, two stemming from the neural stem cell clusters in the earliest time point and one from the gliogenic progenitor cluster in the late embryonic/ early postnatal time points (Fig 4A). The first neural stem cell cluster branches out to give rise to URL and VZ progenitors. The URL progenitors then transition to NTZ neurons, ECNNs, UBCs, postnatal GCPs and granule cells, largely recapitulating the glutamatergic lineage [34]. The VZ progenitors branch from the first neural stem cells clusters and transition to Purkinje neurons, one of the three GABAergic cell types known to arise from VZ progenitors. The

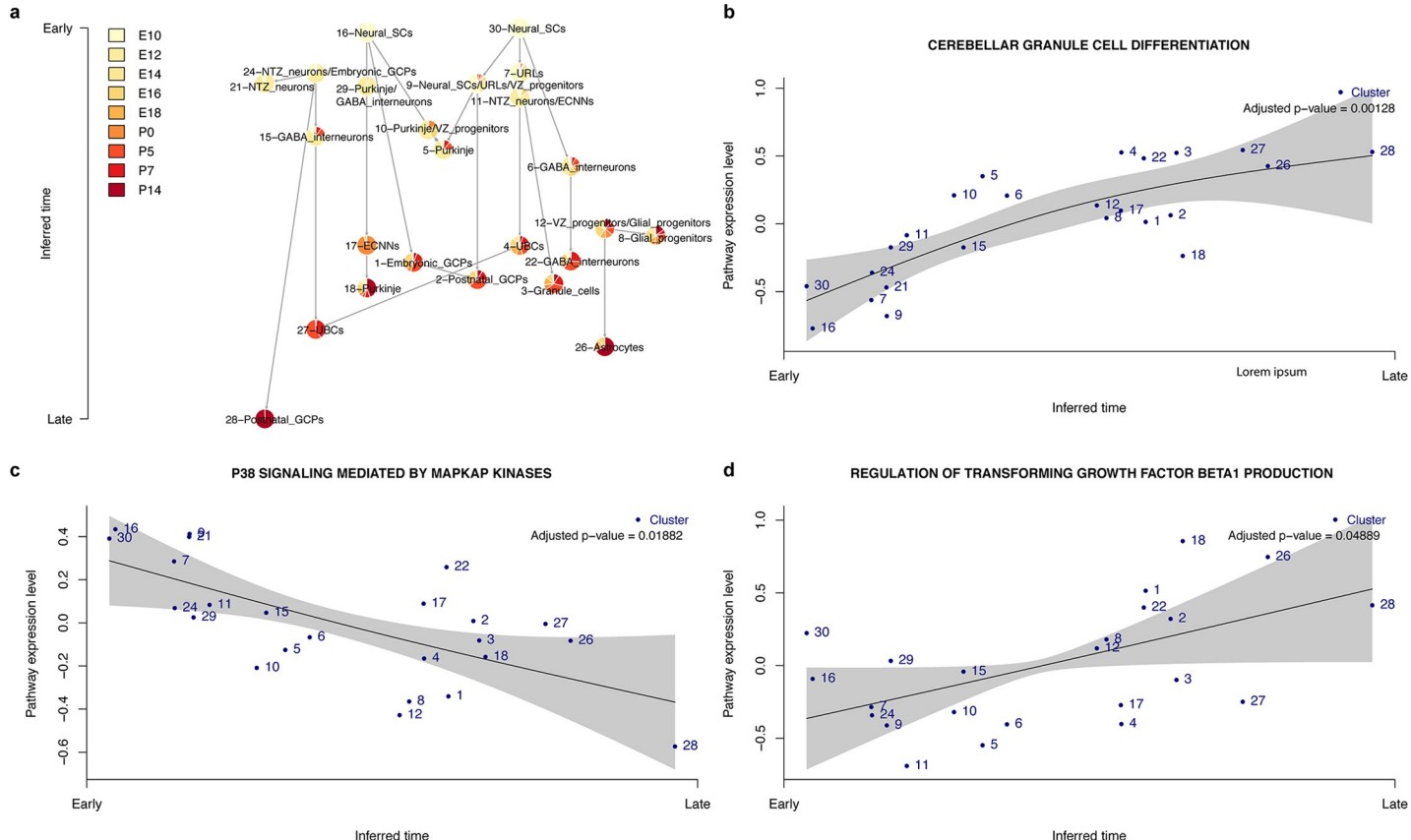

**Fig 4. Tempora analysis of the murine cerebellar data set. a.** Tempora trajectory built on clusters in the murine cerebellar data set. **b–d.** Time-dependent pathways in the murine cerebral cortex data set identified by Tempora's pathway exploration feature. Blue dots represent clusters/cell types, the black line is the best fit of a generalized additive model (GAM) on the pathway enrichment scores of each pathway across all clusters, and grey areas depict 95% confidence intervals of the fitted model.

other neural stem cell cluster also transitions to cell types of both glutamatergic and GABAergic lineages, giving rise to embryonic/postnatal GCPs and GABAergic interneurons, respectively. Finally, Tempora predicts a third lineage of glial cells emerging in late embryonic time points, starting with a VZ progenitors/gliogenic progenitors cluster and branching to astrocytes. This aligns with the understanding that glial cells and astrocytes arise from VZ progenitors but differentiate quite late compared to other cerebellar cells [34]. Overall, Tempora accurately identifies the start and end cell types of the developmental processes and captures the main structures of the trajectories present in the data set.

We identified pathways that change over time using Tempora's pathway exploration feature. Differentiation pathways of multiple cerebellar cell lineages, including GABAergic neurons and granule cells, are enriched later in the time series (Fig 4B), consistent with our understanding that progenitor cells differentiate into terminal cell fates as time progresses [34]. Besides differentiation pathways, pathways related to neuronal activities, such as axonal transport, and signaling pathways, such as TGF-β, are enriched later (Fig 4D). TGF-β's late enrichment in the cerebellar development time series, after the emergence of neurons, aligns with its known role in regulating proliferation and synapse formation in cerebellar neurons [35,36]. Meanwhile, pathways related to stem cell maintenance and signaling pathways such as P38 MAPK are enriched early in the time series (Fig 4C). P38 kinase signaling has been shown

to be essential for the proliferation of cerebellar granule neuron precursors, consistent with its early enrichment in the time series [37].

## Tempora outperforms other trajectory inference methods

To evaluate Tempora's performance and compare it to other cell trajectory inference methods, we measured the ability of a selected set of methods to recapitulate a gold standard set of known cell trajectories we manually collected from the literature for the three developmental processes analyzed above (muscle, neural cortex and cerebellum). For ease of comparison, we formalized all trajectories, both predicted and known, as graphs (networks), with nodes representing cell types, and directed edges representing parent-child relationships between connected nodes. We used two performance scores to assess each method: graph edit distance (GED) or 'mismatch score', which measures the number of edge and node additions or removals required to transform the inferred trajectory to the known trajectory, and F1 score or 'accuracy score', which is the harmonic mean of precision and recall of gold standard directed edge identification.

To assess performance on the human skeletal muscle myoblast (HSMM) data set, which contains 271 cells collected at 0, 24, 48 and 72 hours after the switch of human myoblast culture from growth to differentiation media, we manually curated a corresponding gold standard human myoblast *in vitro* differentiation trajectory through literature search [23,38,39,40] (Fig 5F) and compared Tempora's inferred trajectories to this gold standard. Human myoblasts, after exiting the cell cycle, transition through intermediate states before differentiating into myotubes [23,41]. Since myoblasts have varied differentiating potentials and rates, a portion of them will become myotubes while the rest remain undifferentiated, i.e. they do not, or have yet to, express myogenic transcription factors such as MYOG, which leads to two possible branches from the intermediate state(s) [22]. The starting culture, however, is often contaminated with fibroblast cells, which exert paracrine influence on the differentiation process but cannot differentiate into myotubes [21]. These contaminating cells, thus, form a branch separate from the main differentiation trajectory (Fig 5F).

Tempora's predicted lineage (Fig 2A) closely aligns with the known lineage, except it connects the myotubes cluster to the myoblasts instead of to the intermediate state. This results in a mismatch score of 1, which means that one edge need to be changed in Tempora's output to match the gold standard (Fig 5G). Furthermore, Tempora achieves a high accuracy (F1) score of 0.78 as it is able to infer the correct directions of most edges in the trajectory, except for the missing intermediate state to myotubes connection (Fig 5H). This result demonstrates that Tempora is able to infer a trajectory in the HSMM data set that is mostly consistent with the gold standard.

To assess performance on an embryonic murine cerebral cortex development scRNA-seq data set, which contains approximately 6,000 neural cells collected at embryonic days 11.5 (E11.5), E13.5, E15.5 and E17.5 [5], we manually curated a corresponding gold standard trajectory [5,26,27,28]. Murine corticogenesis consists of transitions between well-characterized cell types. The apical precursors (APs), which delaminate from the neuroepithelium, divide asymmetrically to self-renew and give rise to neurons [26]. At around E11, APs transition to radial precursors (RPs), which continue the asymmetric division to generate neurons either directly or indirectly through IPs [26,27] (Fig 6F). Tempora's inferred trajectory of the murine cerebral cortex data set achieves a low mismatch score and high accuracy score. It predicts almost all known transitions between different cell types in the system, only missing the IPs to neuron connection, which results in a mismatch score of 1 (Fig 6G). Tempora achieves a high accuracy score of 0.9 on the murine cerebral cortex data set, demonstrating that it can accurately identify directed connections between cell types in this larger data set with multiple branches (Fig 6H).

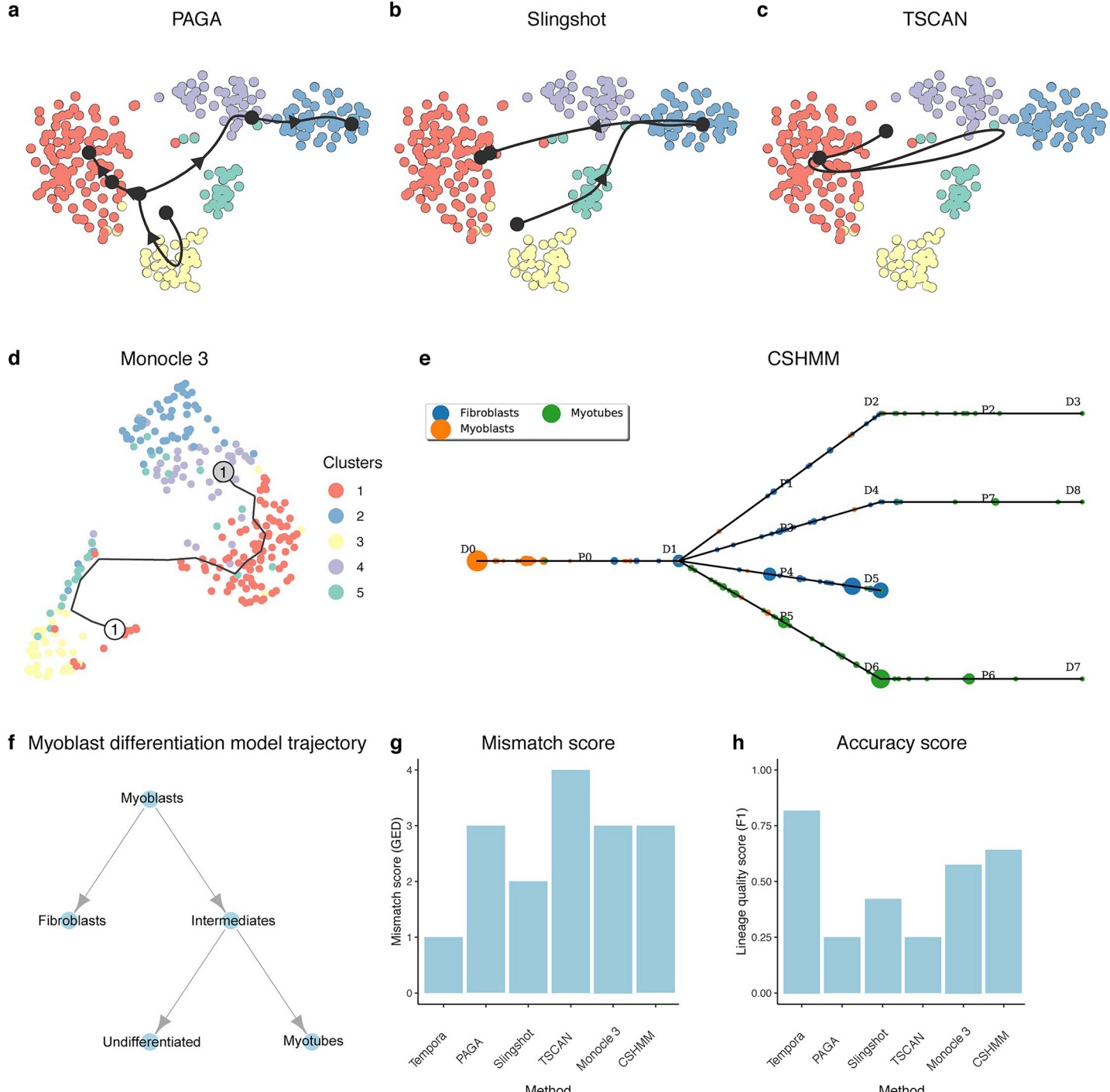

**Fig 5. Performance evaluation on the HSMM data set. a-c.** Trajectories of the HSMM data set inferred by by **a.** PAGA, **b.** Slingshot, **c.** TSCAN, **d.** Monocle 3 and **e.** CSHMM. **f.** The gold standard trajectory used to evaluate the accuracy of all inferred trajectories. **g.** Mismatch scores and **h.** accuracy scores of trajectories from the evaluated methods.

Finally, to assess performance in a much larger and more complex data set, the murine cerebellar development time-series containing ~55,000 neural cells collected at nine time points [34], we collected the curated developmental trajectory made available by the authors (Fig 7E)

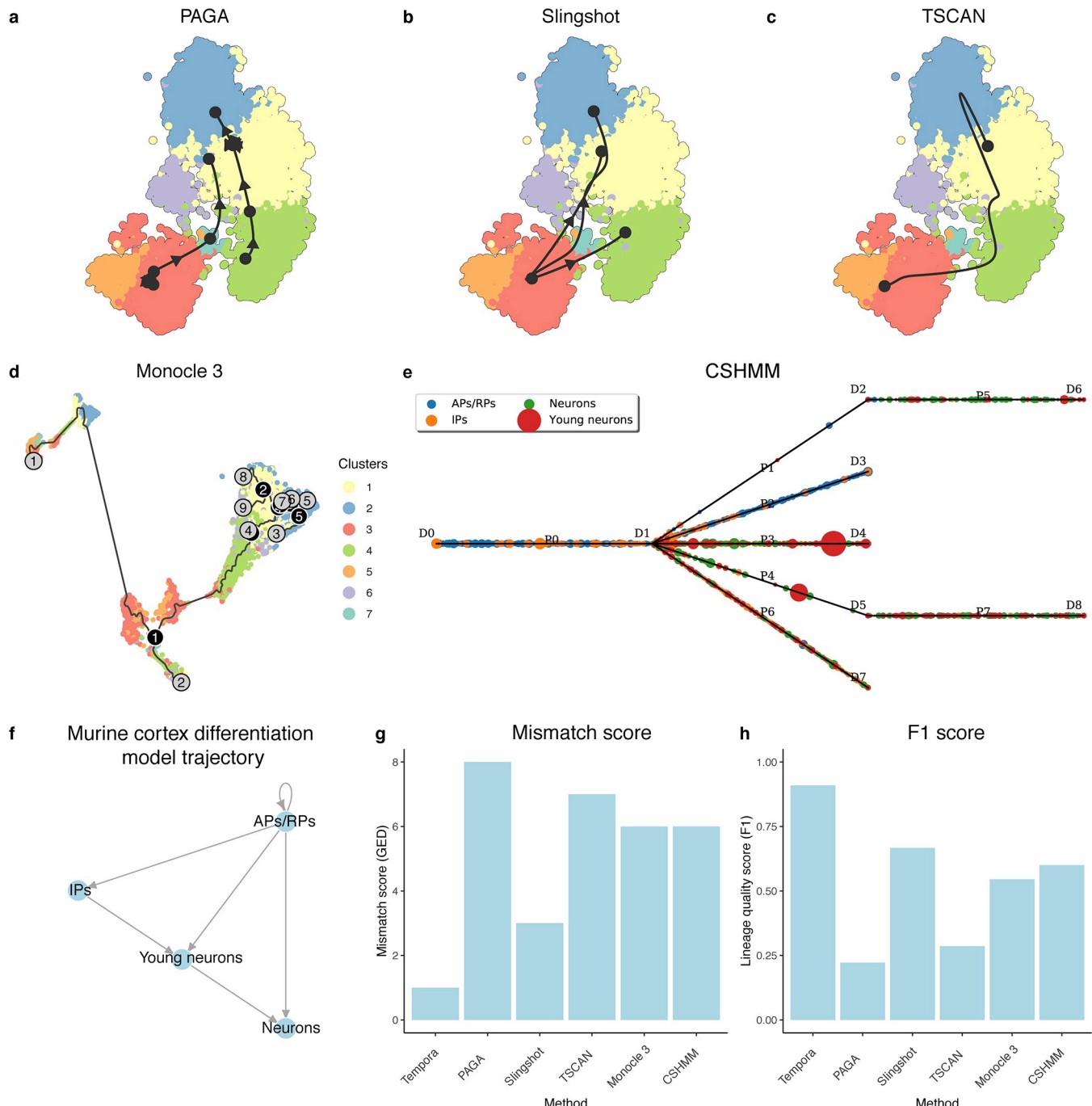

**Fig 6. Performance evaluation on the murine cerebral cortex data set. a-c.** Trajectories of the murine cerebral cortex data set inferred by by **a.** TSCAN, **b.** PAGA, **c.** Slingshot, **d.** Monocle 3 and **e.** CSHMM. **f.** The gold standard trajectory used to evaluate the accuracy of all inferred trajectories. **g.** Mismatch scores and **h.** accuracy scores of trajectories from the evaluated methods.

[34]. Tempora is able to recapitulate the three main lineages present in this data set as well as their timing. Specifically, Tempora predicts that neural stem cells give rise to VZ and URL progenitors, which then differentiate into multiple cell types of the glutamatergic and GABAergic lineages, respectively. However, the high number of cell types leads to mistakes in placing

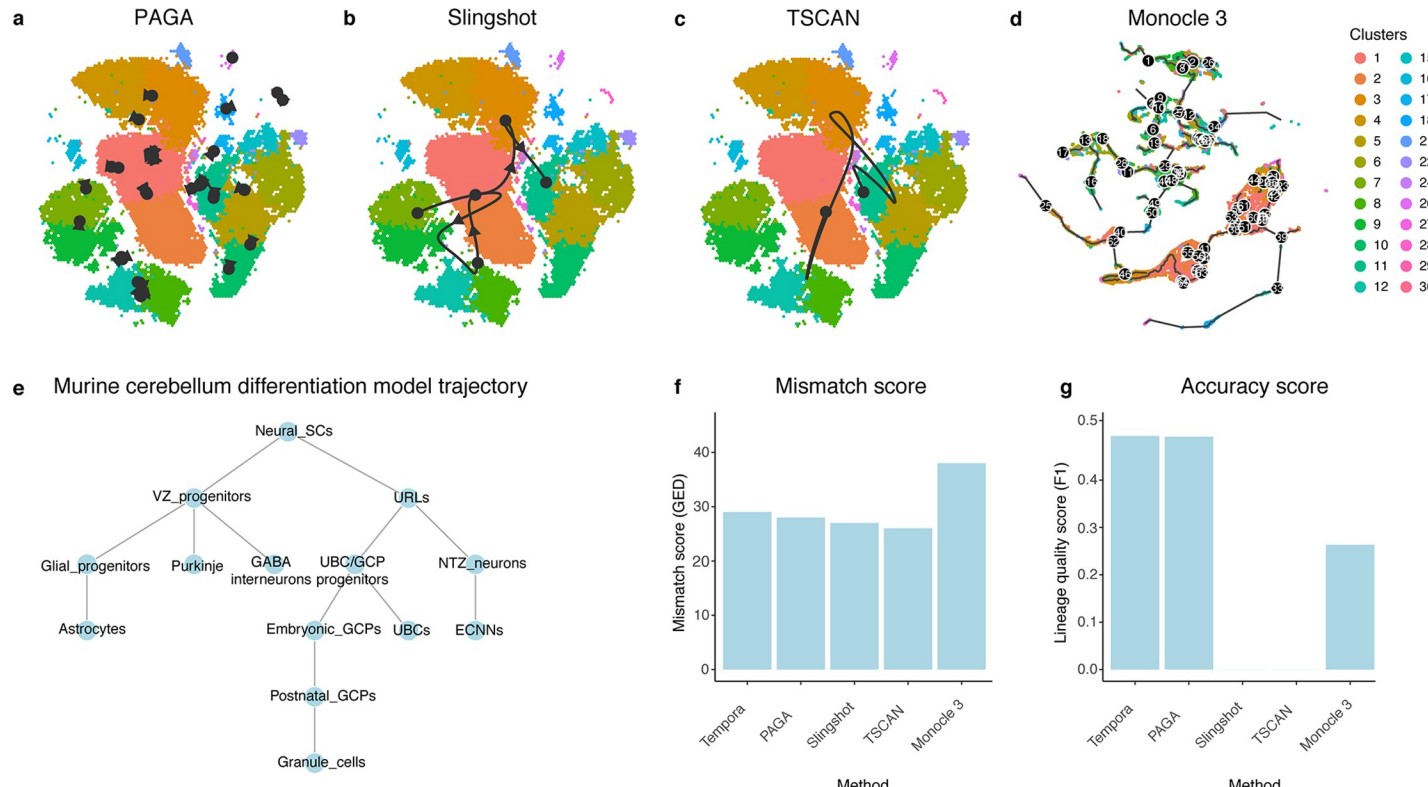

**Fig 7. Performance evaluation on the murine cerebellar data set. a-e.** Trajectories of the murine cerebellar data set inferred by **a.** TSCAN, **b.** PAGA, **c.** Slingshot, **d.** Monocle 3. **e.** The gold standard trajectory used to evaluate the accuracy of all inferred trajectories. **f.** Mismatch scores and **g.** accuracy scores of trajectories from the evaluated methods.

some clusters into their correct lineages. For example, Tempora mistakenly places a cluster of ECNNs, a glutamatergic cell type, in the GABAergic lineage and a cluster of GABA interneurons in the glutamatergic lineage. Finally, Tempora separates the glial lineage as a distinct third lineage instead of connecting it to the VZ progenitor origin. Tempora achieves a mismatch score of 29 and an accuracy score of 0.5.

We next compared Tempora's performance with selected state-of-the-art trajectory inference methods on the validation data sets described above (see Methods). Given the large number of available trajectory inference methods, we chose to compare Tempora against Slingshot [11] and PAGA [13] which are top-performing methods as evaluated by Dynverse [8], as well as TSCAN (cluster-based trajectory inference) [10] and CSHMM (trajectory inference based on temporal information) [15], which includes concepts similar to Tempora (clusters and time point ordering, respectively). We also compared Tempora's performance with Monocle 3, the latest version of the pioneering trajectory inference method Monocle that has been shown to perform well with large data sets [14].

When applied to the HSMM data, Tempora outperforms all other methods (Fig 5). Slingshot and Monocle 3 predict linear trajectories between myoblasts and myocytes, while TSCAN predicts a linear trajectory between fibroblasts and myocytes (Fig 5B–5D). The linear topology of these predicted trajectories does not reflect the separation of the fibroblast population from the muscle lineage. Furthermore, TSCAN's trajectory inaccurately connects two cell types of different lineages (Fig 5C). PAGA and CSHMM both predict branched trajectories (Fig 5A and 5E). PAGA's trajectory starts at the myoblast cluster and branches out to the fibroblasts as

well as the myoblasts, missing the intermediate state (Fig 5A). CSHMM's trajectory incorrectly predicts that myoblasts transitions to fibroblasts before branching out to multiple fibroblast and myoblast terminal states (Fig 5E). Overall, out of all methods that predict branching trajectories, Tempora performs best in terms of mismatch score as it correctly identifies the expected cell states and their directions in the HSMM data (Fig 5G). To calculate the accuracy score on undirected trajectories inferred by other methods, we first determined the origin of the trajectories based on high expression of *CDK1*, *CCND5* for myoblasts in the HSMM data set [9], then added directions to the inferred trajectories by directing all edges outward from the origin. Tempora's accuracy of 0.78 outperforms all other methods, whose accuracy scores range from 0.25 to 0.6 (Fig 5H).

Tempora outperforms all other methods on the murine cerebral cortex data set, which contains more transitions than the HSMM data set (Fig 6). PAGA infers two distinct trajectories, one starting at the AP/RP population and transitioning to IP cells then eventually to neurons, while the other is rooted at the IP population and transitioning to neurons. This inferred trajectory misses the connection between AP/RP and IP cells, as well as omits the young neuron state from the trajectory (Fig 6A). TSCAN predicts a linear trajectory from AP/RP cells to neurons, which miss the multiple transitions in between the earliest and the latest state (Fig 6C). Monocle 3 predicts that a small AP/RP population self-renews, then branches out into two populations, one of which transitions into neurons while the other one remains IPs (Fig 6D). This trajectory predicts most transitions in the data set in the right order, but inaccurately labels IPs as a terminal cell type (Fig 6D). The trajectories that most accurately capture the gold standard are predicted by Slingshot and CSHMM (Fig 6B and 6E). Slingshot predicts a branched trajectory, with APs/RPs branching out to neurons, young neurons and IPs, which captures the main transitions but fails to recapitulate their correct ordering (Fig 6B). CSHMM predicts a mixture of APs/RPs and IPs transitioning to APs/RPs, IPs, young neurons and neurons (Fig 6E). Similar to Slingshot, this trajectory predicts most transitions but misses their sequence. Overall, the performance of the trajectory inference methods on the murine cortex varies greatly, leading to a range of mismatch scores between 3 and 8, all of which are worse than Tempora's score of 2 (Fig 6G). To calculate accuracy scores on trajectories from these methods, we used *Sox2* neural stem cell marker expression to infer the root of the inferred trajectories of all methods except Tempora and CSHMM, and determined that all edges go outward from this root. Tempora (accuracy score of 0.9) outperforms other methods on the murine cerebral cortex data set, whose accuracy scores range from 0.25 to 0.66 (Fig 6H).

Finally, we compared Tempora's performance with other methods when applied to the large murine cerebellum data set (Fig 7). CSHMM was not able to finish running on this data set, even after the number of input genes was restricted to the top 1,000 highly variable genes. TSCAN predicts a linear trajectory connecting an early cluster of glutamatergic NTZ neurons with a cluster of postnatal GCPs, also belonging to the glutamatergic lineage (Fig 7C). This simplistic trajectory omits the majority of transitions between cell types present in the data. Slingshot's predicted trajectory connects URL progenitors to embryonic GCPs, which then transition to VZ and gliogenic progenitors, then to granule cells and finally to NTZ neurons (Fig 7C). This predicted trajectory, even though slightly more complex, erroneously connects cells from multiple lineages together (URL progenitors, embryonic GCPs, granule cells and NTZ neurons from the glutamatergic lineage, VZ progenitors from the GABAergic lineage and gliogenic progenitors from the glial lineage) and the order of transitions does not align with when certain cell types appear (e.g. granule cells emerge after NTZ neurons but are predicted to transition to NTZ neurons by Slingshot). PAGA predicts multiple distinct lineages mostly with connections within cell types, potentially representative of self-renewal or maturity, as well as a lineage including VZ progenitors branching out to gliogenic progenitors and

VZ progenitors. The latter then connect to GABAergic interneurons, Purkinje neurons and NTZ neurons (Fig 7A). The lineage PAGA captures is mostly accurate with our understanding of the GABAergic lineage in the cerebellum, with the exception of VZ progenitors transitioning to glutamatergic NTZ neurons. Finally, Monocle 3 predicts a complex trajectory that spans multiple lineages. It correctly predicts the transition of neural stem cells into URL and VZ progenitors, but subsequently makes multiple errors in organizing the different cell types in the correct lineages. For example, astrocytes are predicted to emerge from UBCs, which actually belong to the glutamatergic lineage (Fig 7D). Therefore, even though Monocle 3's trajectories span all cell types present in the data set, they also include multiple spurious connections that are unlikely to occur biologically. To calculate accuracy scores on trajectories from these methods, we used *Nes* neural stem cell marker expression to infer the root of the trajectories inferred by all other methods, except Tempora and CSHMM, and determined that all edges go outward from this root. Overall, no evaluated method, including Tempora, is able to recapitulate all the complex lineages present in the murine cerebellum data set, but PAGA and Tempora perform best in terms of accuracy score, each obtaining a score of 0.5, which is substantially better than all other methods (ranging from 0 to 0.25, Fig 7F). Tempora, PAGA, Slingshot and TSCAN have similar mismatch scores in the high 20s, while Monocle 3's trajectory results in higher mismatch scores in the 30s (Fig 7G).

## Pathway enrichment information is important for Tempora's performance

To understand the impact of using pathway enrichment profiles on trajectory inference compared to the gene expression profile input used by other methods, we compared trajectories in all three benchmarking data sets using Tempora with and without the pathway enrichment analysis (PEA) step. Removing the PEA step resulted in poorer performance, as evident in an up to 4-fold increase in mismatch scores and a 3-fold decrease in accuracy scores (Fig 8).

Upon closer examination of the resulting trajectories, we observed that gene-input trajectories contain more edges between clusters with similar temporal scores compared to pathway-input trajectories, whose edges often connect clusters from different time points. We propose that this trend can be explained by the high similarity in gene expression profiles of clusters that are closer in developmental time, a fundamental assumption made by trajectory inference methods that rely on gene expression profile-based distance metrics to order cells. To test this hypothesis and better understand the discrepancies in inter-cluster gene vs. pathway enrichment profile similarity, we calculated the Pearson correlation between the gene and pathway enrichment profiles of all pairs of clusters in each data set. We found striking differences in the dynamic range of correlations observed: while correlations between gene expression profiles are uniformly strong and positive across all pairs of clusters, correlations between pathway enrichment profiles are negative for clusters of different cell types (neurons vs. APs, myoblasts vs. fibroblasts) and positive for clusters of the same cell types (neurons vs. neurons) (S4 Fig). These differences suggest that the highly similar gene expression profiles across clusters make them less informative than pathway enrichment profiles in capturing changes along a trajectory.

## Time series information is important for Tempora's performance

Since time is an important component of Tempora, we identified the minimal number of time points required for Tempora and evaluated the effect of time point down sampling on Tempora's performance by re-running Tempora on the HSMM and murine cortex data with cells from one or two time points removed. Tempora's performance, on average, decreased slightly on both data sets when one time point was removed and further decreased when two time

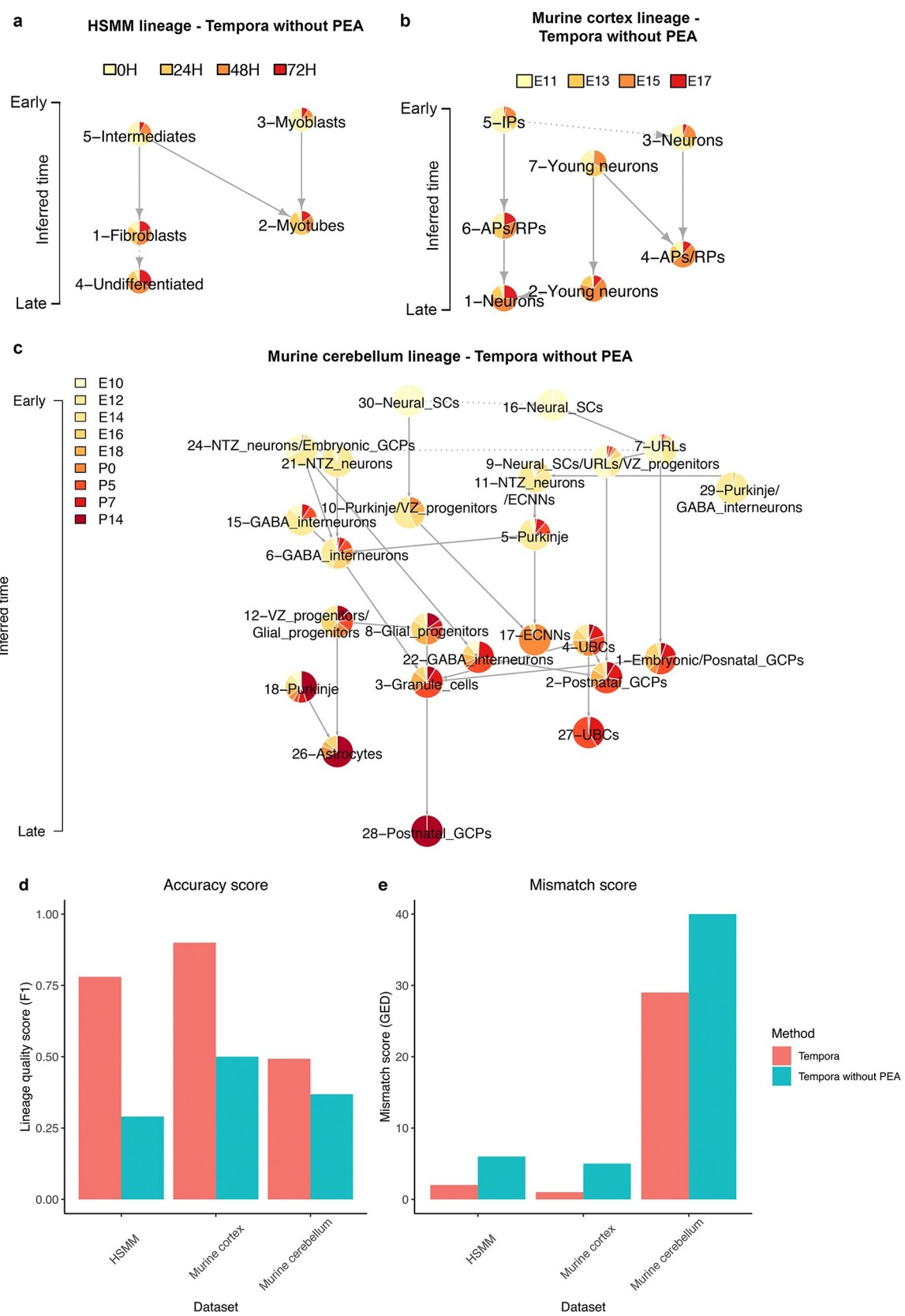

**Fig 8. Performance evaluation of Tempora with and without pathway enrichment analysis (PEA). a-c.** Trajectory inferred by Tempora without PEA of the **a.** HSMM, **b.** murine cerebral cortex and **c.** murine cerebellum data set. **d.** Accuracy scores and **e.** mismatch scores of Tempora trajectories with and without PEA.

points were removed (S10 Fig). Even though this decrease is observed on average, some time points are less important than others, as removing some time points resulted in similar performance compared to when the full set of time points were used (S10 Fig). For example, when cells from the 24 hour time point or from the 24 hour and 72 hour time points were removed from the HSMM data set, the resulting trajectories still contain the majority of the biologically known connections between cell types (S10A and S10B Fig). These results show that Tempora is compatible with data containing as few as two time points, however, including additional time points is beneficial.

We also demonstrated the impact of using available time information in determining trajectory edge directions. Instead of using time ordering information to direct edges, as normally done by Tempora, we determined edge directions in Tempora-inferred trajectories by first identifying the root(s) nodes by expression of a known early marker gene, then directing all edges outwards from the root(s) (see Methods). This is the same approach we used to direct edges and calculate accuracy scores for trajectories inferred by methods that do not support automatic edge direction inference (PAGA, Slingshot, TSCAN and Monocle 3). Determining directions with marker genes does not lead to any changes in the HSMM trajectory, as all clusters are connected to the root myoblast cluster (S11A Fig). However, for more complex trajectories in which there is more than one root state, including the murine cortex and cerebellum data sets, Tempora's accuracy scores decrease when directions are determined without using time ordering as edges not connected to the roots become bidirectional (S11B–S11D Fig). Thus, incorporating time point order information in trajectory inference enables a more accurate determination of trajectory edge direction compared to traditional marker gene-based direction inference, especially for larger data sets.

## Discussion

We have described and evaluated Tempora, a novel pathway-based cell trajectory inference method for time-series scRNA-seq data. Tempora uses an information theoretic approach to build a trajectory at the single cell cluster (or cell type) level based on the clusters' pathway enrichment profiles, effectively connecting related cell types and states across multiple time points. Taking advantage of the available time information, Tempora infers the directions of all connections in a trajectory that go from early to late clusters. Evaluation on a diverse set of three time series scRNA-seq data sets with known developmental trajectories demonstrates that Tempora can accurately predict known trajectories, outperforming leading trajectory inference methods. Furthermore, downstream analyses using Tempora's pathway exploration feature identifies pathways known to be important during the process under study as well as additional interesting pathways, demonstrating the method's ability to recapitulate and discover relevant biological signals during development processes.

Tempora constructs a trajectory at the cluster level, instead of at the single cell level. While scRNA-seq offers the opportunity to identify the transitions cells undergo during a dynamic process at cellular resolution, the granularity of a trajectory inferred at the single-cell level can often render it difficult to interpret, especially for larger data sets. Furthermore, current scRNA-seq technologies typically have low gene expression sensitivity at the single cell level, which is a challenge for analysis methods. By analyzing scRNA-seq data at the cluster level instead of at the single-cell level, Tempora: 1) amplifies gene expression signals from one cell

using similar cells in a cluster; 2) eases interpretation, because it operates on a relatively small number of clusters which usually represent known cell types and which naturally fits with the standard practice of clustering scRNA-seq data to identify cell types; and 3) reduces computational time and resources needed to analyze large data sets because it works with a relatively small number of clusters instead of potentially a very large number of cells. Tempora scales linearly with the number of genes and cells (S12 Fig) and takes on average 60 seconds to complete when applied on the ~19,000 gene x ~55,000 cell gene murine cerebellar development expression matrix, while Monocle 3 takes 1700 seconds on a personal computer (MacBook Pro with 2.3 GHz Intel Core i9 processor and 16GB RAM). Other methods evaluated here either ran out of memory (PAGA, Slingshot and TSCAN) or never finished (CSHMM) on this machine. PAGA, Slingshot and TSCAN required a more powerful server with 120GB of RAM to complete and CSHMM never finished even on this machine after spending 180 hours of compute time. However, a key drawback of Tempora's cluster-based approach is that it does not allow users to analyze gradual gene expression profile changes that occur when cells are transitioning types or states. Since other time series based trajectory inference methods, such as Waddington-OT [16], focus on analyzing these gradients, Tempora can be considered a complementary approach and multiple methods should be used to study a dynamic cellular process.

Our analysis follows established scRNA-seq analysis workflows, in which certain decisions can affect Tempora's output. Tempora assumes that user input includes an optimized clustering solution for their data. If the clustering is not optimal, the output trajectory may be too general or too detailed. Challenges regarding clustering are inherent to high-dimensional scRNA-seq data, which relies on user-input parameters to determine the number of clusters, and no gold standard exists to guide the selection of these parameters [42]. Over-clustering, when clusters are split too much (e.g. splitting a single cell type into two clusters), can lead to parallel edges originating from oversplit clusters and terminating at another cluster, ostensibly suggesting a multiple-parent lineage (S5 Fig). Ideally users will make use of their biological expertise to tune their clusters and make them biologically relevant. However, for understudied systems or to consider complementary information, users can detect that a data set is over-clustered when differential gene expression analysis between at least a pair of clusters in their data returns no differentially expressed gene, suggesting that the two clusters are transcriptionally similar and thus should be merged into one. This measure is implemented in the scClustViz software, which we use in this work [43]. On the other hand, under-clustering can result in overly simplified lineages (S5 Fig). Under-clustering can also lead to certain cell types appearing at one time point but absent from another, because they have been clustered with other cell types at one time point and not the other. If a major expected cell type is not represented with a cluster after clustering and annotation, while its marker gene is expressed in the data, it is likely that the data set has been under-clustered and the missing cell type is merged in a large cluster with another similar cell type. In this case, users should re-cluster their data with greater resolution to ensure that all expected cell types are detected.

As time-series data are often collected in batches, technical batch effects are important to consider. We used Harmony to correct for batch effect in the three scRNA-seq data sets used in this study before downstream trajectory analysis. Without such correction, Tempora's performance decreased slightly on all gold standards (S6–S8 Figs). This is likely due to the suboptimal clustering driven by batch effects, which results in less accurate inference of trajectories based on the resulting clusters. We also tested batch correction with Monocle 3, which integrates the Batchelor [44] batch correction method as an optional feature (S9 Fig). Batch correction dramatically improved Monocle 3 performance on the muscle data (HSMM), making it comparable to Tempora, but only slightly on the cerebral cortex data, and reduced performance on the murine cerebellar data (S9 Fig). Other methods evaluated here do not have

integrated data batch correction or do not naturally take Harmony-corrected principal components as input. Overall, we recommend applying batch correction to input data before Tempora analysis, but note that this should be evaluated more broadly in future work to support a more confident and general claim.

Tempora determines trajectory edge direction under the assumption that all differentiation processes occur unidirectionally, so cells from earlier time points give rise to cells from later time points. However, this assumption may fail in certain contexts, such as dedifferentiation of cancer cells to a more stem-like state [45]. Tempora would still be able to infer trajectories in these contexts; however, users should be cautious with interpreting the results, as cells in later time points may not be in a more differentiated state compared to cells in earlier time points.

Tempora depends on pathway information, which we found to help distinguish clusters from each other compared to gene expression profiles. Pathway enrichment of gene expression profiles increases sensitivity and statistical power, as well as reduces noise [46,47]. We interpret our results to indicate that these advantageous properties of pathway information are the reason why using pathway information in Tempora performs better than using gene expression profiles at correctly identifying cell cluster/type transitions. While we have made an effort to collect a comprehensive set of pathway information, pathway information is incomplete and many genes are not covered. If there is low pathway coverage of a particular system under study or if data bias (e.g. gene drop out) correlates with pathway structure, it could negatively affect Tempora performance.

Time series scRNA-seq is increasingly used to investigate dynamic biological processes, including development and differentiation. Compared to single snapshot experiments, it offers advantages in that more cells and types are captured that may only be present at certain time points, and that cell transition processes are more directly measured. However, many computational challenges exist with this data, including how to track changing cell types and states between time points [48]. Time point ordering information is useful to supervise trajectory inference and enables accurate identification of cell types consisting of cells from different time points as well as lineage connections between them. When combined with biological pathway information, time-series based analysis can generate useful insights into dynamic processes. Future work to integrate the Tempora framework with other time-series scRNAseq analysis concepts, including dimensionality reduction with TSEE [49] and population dynamics inference with pseudodynamics [50], may further improve Tempora performance.

## Methods

### Single-cell RNAseq data

Three time-course scRNA-seq data sets were used to validate Tempora. The first data set consists of cells at four time points during the *in vitro* differentiation of human skeletal muscle myoblasts (HSMM). Read count for this data set was accessed from GEO, accession number GSE52529 [9]. The second data set was sampled at four time points during early murine cerebral cortex development. This data set was downloaded from GEO, accession number GSE107122 [5]. The third data set consists of cells at nine time points during murine cerebellar development and was accessed from GEO, accession number GSE118068 [34].

All data sets were filtered to remove low expressed genes (defined as those found in less than 3 cells) and damaged cells with high mitochondrial genome transcript content (4 median absolute deviations above the median). After this initial filtering step, the murine cerebral cortex data were further filtered to remove non-cortical cells, as done in the original publication [5]. These included cells expressing *Aif1* (microglia), hemoglobin genes (blood cells), collagen genes (mesenchymal cells), as well as *Dlx* transcription factors and/or interneuron genes

(ganglionic eminence-derived cells) [5]. The murine cerebellar data set was also filtered to remove non-neural cells, including mesenchymal stem cells (expressing *Prrx2*), brainstem progenitors (expressing *Olig3*), endothelial cells (expressing *Cldn5*), blood cells (expressing hemoglobin genes), meninges (expressing *Cxcl12*), pericytes (expressing *Rgs5*) and microglia (expressing *Aif1*). The data sets were then normalized using the deconvolution method implemented in the *scran* R package, which pools cells with similar gene expression profiles and library sizes together to normalize [51]. Afterwards, cells were iteratively clustered in Seurat at increasing resolutions until the number of differentially expressed genes between two neighboring clusters reached 0, as determined by scClustViz [43]. We then chose the optimal clustering resolution, defined as the point where the number of clusters was maximized while the number of DE genes between neighboring clusters remained larger than 0. Cell clusters for all data sets were annotated using GSVA [17] with known marker genes for expected cell types published with each data set (S1 Table). Specifically, a pseudo-bulk average expression profile was determined for each cluster and input into GSVA to evaluate for enrichment in marker gene sets as recommended in [52]. For each cluster, the cell type that has the highest score (or cell types in case of tie in scores) output by GSVA is used as the cluster label. The resulting annotated clusters represent cell types or states that are stable over the developmental process, such as apical progenitor cells in murine cerebral cortex development and myoblasts in muscle development. These annotations were performed once and used for all methods and analyses to ensure consistent comparisons.

## Data preprocessing, batch effect correction and clustering

Tempora takes processed scRNA-seq data as input, either as a gene expression matrix with separate time and cluster labels for all cells, or a Seurat object containing gene expression data and a clustering result. Tempora does not implement clustering or batch effect correction as part of its pipeline and assumes that the user has input a well-annotated cluster solution free of batch effect into the method.

Since a good clustering result is important to the successful application of Tempora on a data set, we recommend users take advantage of methods such as scClustViz [43] to visualize clusters at different resolutions, analyze cluster relationships across resolutions as well as investigate marker gene expression to help choose appropriate clustering parameters [53].

## Pathway enrichment analysis

Tempora calculates the average gene expression over all cells in a cluster for all clusters as input by the users and determines the pathway enrichment profile of each cluster using GSVA [17]. By analyzing scRNA-seq data on the cluster level instead of the single-cell level, Tempora amplifies gene expression signals from similar cells in a cluster to alleviate the typical problem of low sensitivity per cell of popular scRNA-seq experimental methods, as well as to reduce the number of nodes in the inferred lineage, reducing computational time and allowing users to interpret the lineage more easily. The default pathway gene set database Tempora uses is the Bader Lab pathway gene set database without electronic annotation Gene Ontology terms (in this work, version Human_GO_AllPathways_no_GO_iea_August_01_2019_symbol.gmt was used, accessed at http://download.baderlab.org/EM_Genesets/current_release/Human/ symbol/Human_GO_AllPathways_no_GO_iea_August_01_2019_symbol.gmt, and Mouse_ GO_AllPathways_no_GO_iea_August_01_2019_symbol.gmt, accessed at http://download. baderlab.org/EM_Genesets/current_release/Mouse/symbol/Mouse_GO_AllPathways_no_ GO_iea_August_01_2019_symbol.gmt), filtered to include gene sets between 10 and 500 genes in size [47]. The enrichment scores of all P pathways in each cluster make up the cluster's

pathway enrichment profile, which is a vector of length P. Since pathway gene set databases contain redundant pathways and this redundancy is not evenly distributed across the database (e.g. well studied pathways are better represented), Tempora uses PCA to reduce redundancy in all pathway enrichment profiles and identifies the top $n$ principal components, defined as the components before the slope levels off ("the elbow"), in a scree plot, to input to downstream trajectory construction steps. In this study, 5 PCs were used for analysis of the HSMM data, 6 for the murine cortex data and 8 for the murine cerebellar data. This is a user definable parameter.

## Filtered mutual information network

We conceptualize the cell (cluster) trajectory from different time points of a developmental process as a graph (network), where vertices represent clusters and edges represent parent-child relationships between these vertices. Tempora employs the gene expression rank-based mutual information (MI) approach implemented in ARACNE [19] to calculate MI between all cluster pairs present in the data. The data-processing inequality is then applied by ARACNE to remove the edge with the lowest MI in each triple to reduce the number of indirect interactions between clusters. This results an undirected network where nodes are cell clusters and weighted edges represent MI strength relationships between clusters. All ARACNE edges are kept for Tempora analysis (no threshold used).

## Direction identification

Tempora uses time information to determine the edge directions in the constructed MI network. Tempora assigns each time point a sequential, ordinal value corresponding with its distance from the earliest time point and calculates a temporal score for each cluster based on its composition of cells from each time point. Specifically, the temporal score, $T_k$, of cluster $k$ consisting of $p_i$ percent cells at time point $i$ is calculated as:

$$T_k = \sum_{i=1}^{N} p_i \cdot i$$

Where $N$ is the number of time points. Under the assumption that cell differentiation progresses unidirectionally from stem or progenitor cells (early time points) to differentiated (late time points) cells, Tempora assigns directions to all edges in the network so that edges point from clusters with low temporal scores to clusters with high temporal scores. For edges that connect clusters with similar temporal scores (with the similarity threshold defined by the users, default used here = 0.01), Tempora does not assign directions as these edges can represent small transitions in cell states over a short time, in which the unidirectional assumption may not hold. These undirected edges are visualized as dashed lines in output trajectories and count as a bidirectional edge in performance comparisons. The 0.01 threshold was optimized based on the data sets analyzed here.

## Identification of time-dependent pathways

Tempora identifies pathways that vary over time by fitting a generalized additive model on the pathway enrichment scores of each pathway across all clusters over all time points and uses ANOVA to compare the fitted model with the null model of uniform pathway enrichment over all time points. Pathways with adjusted p-values below a user-defined threshold, with a default value of 0.05, are reported as significantly varying over time. The model fitting and statistical testing are done using the *mgcv* package in R.

## Evaluation and comparison with other methods

We evaluated Tempora by comparing its predicted cell lineages to known lineages manually curated from the literature. The lineages are represented as directed acyclic graphs, with vertices representing cell types and directed edges representing lineage connections. Two approaches were used to measure the accuracy of Tempora and other methods in predicting the known lineage: a mismatch score, implemented as the graph edit distance (GED), and accuracy, implemented as the F1 score.

**Model trajectory construction.** We manually curated the model trajectories for the *in vitro* differentiation of human myoblasts and murine cortical development through literature search [23,26,27,28,38,39,40]. The gold standard trajectory for the murine cerebellum data set is taken from the original publication [34]. All gold standards are established at cell population (cluster) level. We represented the lineage relationships between different cell types in a system as a directed acyclic graph. Each node in a model trajectory represents a distinct cell type as noted in the literature and described in the Cell Ontology [54], while the edges represent lineage connections (*develops_from* relationship in the Cell Ontology) between these cell types.

**Mismatch (GED) score.** We used the unweighted GED metric to measure the number of mismatches between the predicted and known trajectory, both formalized as undirected graphs to enable comparisons with methods that do not predict edge directions [55]. GED is formally defined as the smallest total number of graph edit operations needed to transform one graph into another. In this context, the permitted operations included insertion and deletion of edges or vertices.

To calculate the mismatch score between a pair of graphs, we first take the cell cluster labelling outputs from GSVA [17], which we use as an automated cluster labeling process using known marker gene sets for expected cell types in a data set taken from the original scRNA-seq publication. We then calculate the number of differences in the cell types of the predicted and known trajectories, as well as in the adjacency matrices of both trajectories. The sum of these two differences is the mismatch score for each pair of graphs.

**Accuracy (F1) score.** To compare the accuracy of Tempora's time-based direction inference with the model trajectory, we calculated the F1 score on each predicted trajectory as follows:

$$F_1 = 2 \cdot \frac{precision \cdot recall}{precision + recall}$$

in which true positives (TP) are edges present in both the model and the predicted trajectory, false positives (FP) are edges in the predicted trajectory but not in the model, and false negatives (FN) are edges in the model but not in the predicted trajectory. An edge in the predicted graph is considered true positive only when its two vertices and direction match those of an edge in the model graph.

To calculate the accuracy score on undirected trajectories inferred by Monocle 2 and TSCAN, we first determined the origin of the trajectories based on high expression of a set of known early marker genes (*CDK1*, *CCND5* for myoblasts in the HSMM data set [9], *Sox2* for apical precursors in the murine cerebral cortex data set [5] and *Nes* for neural stem cells in the murine cerebellar data set [34]), then added directions to the inferred trajectories by directing all edges to go outward from the origin.

**Monocle 3.** We applied Monocle 3 [14] on the validation data sets in this study using the method's recommended protocol. For each data set, the full normalized gene expression matrix and meta data was used to construct the cell_data_set object for Monocle 3 analysis. The data was then preprocessed to reduce its dimensionality using PCA. As the gene

expression matrix we input was already normalized, it was not further log-normalized in the preprocessing step. Following PCA, UMAP was applied on the data to further reduce its dimensionality. The parameters for this step were set as Monocle 3's default (umap.metric = "cosine", umap.min_dist = 0.1, umap.n_neighbors = 15, umap.fast_sgd = FALSE, umap. nn_method = "annoy"). Cells in the data set were then clustered with the Louvain/Leiden community detection method based on the PCA reduced dimensionality space. The parameters for this step were set as Monocle 3's default (k = 20, cluster_method = "leiden", num_ iter = 2, partition_qval = 0.05, weight = FALSE). The fine-grained principal graph describing potential paths cells could take through development was then constructed with the learn_- graph() function, keeping all parameters as default. We then determined the root of the trajectory by marker gene expression as described in the section on accuracy score calculation above and used this information to order the cells in pseudotime.

To compare Monocle's performance with and without batch effect removal, we additionally corrected any batch effect using Monocle 3's align_cds() function after preprocessing the data and before dimensionality reduction with UMAP. The downstream steps after align_cds() are the same for aligned and non-aligned data.

To formalize a Monocle trajectory as a graph, we considered each state, or segment of the tree, as a vertex, and connected the vertices with appropriate edges to recapitulate Monocle's output.

**TSCAN.**   We applied TSCAN [10] on the validation data sets in this study using Dynverse's wrapper, which takes the gene expression matrix of each data set as input and the Tempora clustering solution of cells in each data set as an optional prior. The clusternum parameter, whose domain is [2,20], was left at the default of [2,9] for the HSMM and murine cortex data set, and set as [2,20] for the murine cerebellum data set to allow for more clusters to be discovered in this large data set. Other input parameters to TSCAN were kept as Dynverse's default, including minexpr_value as 0, minexpr_percent as 0, cvcutoff as 0, cvcutoff as 0 and modelNames as VVV.

**Slingshot.**   We applied Slingshot [11] on the validation data sets in this study using Dynverse's wrapper, which takes the gene expression matrix of each data set as input and the list of start and end cells, as well as the Tempora clustering solution of cells in each data set as optional priors. For each data set, cells from the earliest time point were designated as start cells and cells from the latest time point as end cells. All parameters were kept as Dynverse's defaults (cluster_method = "pam", ndim = 20, shrink = 1, reweight = TRUE, reassign = TRUE, thresh = 0.001, maxit = 10, stretch = 2, smoother ="smooth.spline", shrink.method = "cosine").

**PAGA.**   We applied PAGA on the validation data sets in this study using Dynverse's wrapper, which takes the gene expression matrix of each data set as input and the list of start cells as well as the Tempora clustering solution of cells in each data set as an optional prior. The embedding_type parameter was set as 'umap' for the murine cerebellum data set to accommodate for its larger size and 'fa' for the other two validation data sets. The connectivity cutoff for each data set was determined by running PAGA on the data set with different cutoff between 0.05 and 0.6 at 0.1 increments and choosing the resulting trajectory with the highest accuracy and lowest mismatch score. This cutoff is 0.3 for the HSMM and murine cerebral cortex data set and 0.4 for the murine cerebellar data set. The rest of the parameters were left as Dynverse's default (filter_feature = TRUE, n_neighbors = 15, n_comps = 50, n_dcs = 40, resolution = 1).

**Dynverse trajectory graph formalization.**   We formalized the trajectories output by Dynverse as graphs by taking advantage of Dynverse's milestone network data structure, which describes connection between milestones in a lineage. The milestones are the vertices and the connections between them are the edges in our graph formalization. To determine what

cluster lies at each milestone, we identified cells that are closest to each milestone (highest milestone percentage) and the cluster(s) they belong to according to our original clustering resolution of a given data set. We then determined the roots and directions of the graphs as described in the section on accuracy score calculation.

Trajectories obtained from Dynverse's wrapper methods are plotted in a tSNE reduced dimensionality space using Dynverse's plot_dimred() function. Cells are colored with their cluster assignments, which are the same cluster solutions input to Tempora.

**CSHMM.**   We applied CSHMM [15] on the validation data sets using the method's recommended protocol. The full normalized gene expression matrix of each data set, along with the cells' collection times and the cell type labels of the clusters they belong to, as determined automatically by GSVA (described under the Single-cell RNAseq data section of Methods), was input to initialize scDiff. Afterwards, we trained the CSHMM model for each data set in five iterations and selected the model with the highest accuracy and lowest mismatch score.

## Supporting information

**S1 Fig. The HSMM data set. a.** tSNE plot showing 271 cells in the HSMM data set, colored by cluster number. **b-d.** Visualization of known marker genes for **b.** myoblasts, **c.** myotubes and **d.** fibroblasts on the HSMM data set.
(PDF)

**S2 Fig. The murine cerebral cortex data set. a.** tSNE plot showing the ~6,000 neural cells captured in the murine cerebral cortex data set, colored by cluster number. **b-f.** Visualization of known marker genes for **b.** apical precursors (APs), **c.** radial precursors (RPs), **d.** cycling apical/radial precursors (AP/RPs), **e.** intermediate progenitors (IPs), **f.** early neurons and **g.** neurons in the murine cerebral cortex data set.
(PDF)

**S3 Fig. The murine cerebellum data set. a.** tSNE plot showing the ~55,000 neural cells captured in the murine cerebellum data set, colored by cluster number. **b-i.** Visualization of known marker genes for **b.** neural stem cells. **c.** glutamatergic cells, **d-f.** GABAergic cells, **g-i.** glias in the murine cerebellum data set.
(PDF)

**S4 Fig. Correlation between gene expression and pathway enrichment.** Correlation plots showing cluster-average gene expression and pathway enrichment profiles in **a.** HSMM and **b.** murine cerebral cortex data.
(PDF)

**S5 Fig. The effects of sub-optimal clustering resolution choice on trajectory inference.** Over clustering (middle) can lead to complex lineages with converging connections, while under clustering (right) can lead to oversimplified lineages.
(PDF)

**S6 Fig. Effect of data alignment on Tempora performance on HSMM data. a-b.** tSNE plots of HSMM data **a.** with and **b.** without Harmony alignment, with cells colored by time points. **c.** tSNE plot of clusters in HSMM data without alignment. **d.** Tempora trajectory and **e-f.** performance evaluation of Tempora on HSMM data without alignment.
(PDF)

**S7 Fig. Effect of data alignment on Tempora performance on murine cerebral cortex data. a-b.** tSNE plots of murine cerebral cortex data **a.** with and **b.** without Harmony alignment,

with cells colored by time points. **c.** tSNE plot of clusters in murine cerebral cortex data without alignment. **d.** Tempora trajectory and **e-f.** performance evaluation of Tempora on murine cerebral cortex data without alignment.
(PDF)

**S8 Fig. Effect of data alignment on Tempora performance on murine cerebellar data. a-b.** tSNE plots of murine cerebellum data **a.** with and **b.** without Harmony alignment, with cells colored by time points. **c.** tSNE plot of clusters in murine cerebral cortex data without alignment. **d.** Tempora trajectory and **e-f.** performance evaluation of Tempora on murine cerebellar data without alignment.
(PDF)

**S9 Fig. Effect of batch effect correction on Monocle 3 performance. a, c, e.** Monocle 3 trajectories of **a.** HSMM, **c.** murine cerebral cortex and **e.** murine cerebellar data sets without batch correction. **b, d, f.** Monocle 3 trajectories of **b.** HSMM, **d.** murine cerebral cortex and **f.** murine cerebellar data sets with Batchelor batch correction. **g-h.** Performance evaluation of Monocle 3 on the benchmarking data sets with and without batch effect correction.
(PDF)

**S10 Fig. Effect of time point down sampling on Tempora performance on HSMM and murine cerebral cortex data. a-b.** Tempora trajectory of HSMM data when cells from **a.** 24 hours and **b.** 24 hours and 72 hours are removed. **c.** Mismatch score and **d.** accuracy score evaluation of Tempora performance on the HSMM data set when time points are down sampled. **e-f.** Tempora trajectory of murine cerebral cortex data when cells from **e.** E13 and **f.** E15 and E17 are removed. **g.** Mismatch score and **h.** accuracy score evaluation of Tempora performance on the murine cerebral cortex data set when time points are down sampled. Scores represent an average of four experiments, in which all cells from a different time point or combination of two time points are removed before running Tempora.
(PDF)

**S11 Fig. Effect of time removal on direction determination of Tempora-inferred trajectory. a-c.** Tempora trajectories of **a.** HSMM, **b.** murine cerebral cortex and **c.** murine cerebellum data set, with edge directions determined by identifying the root state(s) with known early marker genes (*CDK1*, *CCND5* for myoblasts in the HSMM data set, *Sox2* for apical precursors in the murine cerebral cortex data set and *Nes* for neural stem cells in the murine cerebellar data set) and directing all edges outwards from the root states. **d.** Accuracy score of Tempora trajectories with edge directions determined without time information.
(PDF)

**S12 Fig. Tempora's runtime scales with the number of cells and genes.** Runtime of Tempora when applied to **a-b.** murine cortex and **c-d.** murine cerebellum data set after downsampling of **a,c.** cells and **b, d.** genes.
(PDF)

**S1 Table. Marker genes used to annotate cell types.**
(PDF)

## Acknowledgments

We thank M. Abou Chakra and B. Innes for helpful discussion and feedback on this manuscript.

## Author Contributions

**Conceptualization:** Thinh N. Tran, Gary D. Bader.

**Data curation:** Thinh N. Tran, Gary D. Bader.

**Formal analysis:** Thinh N. Tran.

**Funding acquisition:** Gary D. Bader.

**Investigation:** Thinh N. Tran, Gary D. Bader.

**Methodology:** Thinh N. Tran, Gary D. Bader.

**Project administration:** Thinh N. Tran, Gary D. Bader.

**Resources:** Thinh N. Tran, Gary D. Bader.

**Software:** Thinh N. Tran.

**Supervision:** Gary D. Bader.

**Validation:** Thinh N. Tran, Gary D. Bader.

**Visualization:** Thinh N. Tran.

**Writing – original draft:** Thinh N. Tran, Gary D. Bader.

**Writing – review & editing:** Thinh N. Tran, Gary D. Bader.

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
