## [Decision Letter · Decision Letter 0]

8 Jan 2020

Dear Dr Tran,

Thank you very much for submitting your manuscript 'Tempora: cell trajectory inference using time-series single-cell RNA sequencing data' for review by PLOS Computational Biology. Your manuscript has been fully evaluated by the PLOS Computational Biology editorial team and in this case also by independent peer reviewers. The reviewers appreciated the attention to an important problem, but raised some substantial concerns about the manuscript as it currently stands. While your manuscript cannot be accepted in its present form, we are willing to consider a revised version in which the issues raised by the reviewers have been adequately addressed. We cannot, of course, promise publication at that time.

Sincerely,

Qing Nie

Associate Editor

PLOS Computational Biology

Thomas Lengauer

Methods Editor

PLOS Computational Biology

[LINK]

Reviewer's Responses to Questions

**Comments to the Authors:**

Reviewer #1: Executive Summary:

Although single-cell RNA-seq (scRNA-seq) provides a singular snapshot in time, sequencing a representative population of cells within a dynamic and developmental process can yield snapshots representing a range of dynamic and developmental stages that can then be ordered in pseudotime and within trajectories. Thus, to infer this pseudotime ordering of cells within putative trajectories by taking advantage of the asynchronous nature of developmental processes, a number of computational trajectory inference methods have been developed. Here, Tran and Bader present a trajectory inference method called Tempora, specifically for time-series scRNA-seq data that can order populations of cells within trajectories while taking into consideration the true time information introduced by the time series. While the ability to infer trajectories from time-series scRNA-seq data is an important and much needed computational methodological development, the rigor of benchmarks and extent of comparisons to existing methods presented here need substantial improvement in order to substantiate conclusions, thus precluding publication at this time.

---

Major Comments:

1. It remains unclear how many time series points are necessary for Tempora. Will Tempora be compatible and robust when only two or three time points are available? Are the inferred trajectories for HSMM and murine cortex stable to downsampling of time points?

2. A number of unified single-cell analysis approaches have recently been published to accommodate joint embeddings across batches of data such as time series data, including Harmony (Korsunsky et al, Nature Methods 2019) and BBKNN (Polanski et al Bioinformatics 2019) to name a few. Indeed, the authors use Harmony corrected PCs prior to Tempora to ensure that trajectories are not driven by batch-effects. It is currently unclear from the Methods section whether similar batch-correction treatments were applied in the Monocle and TSCAN comparison, such that inferior performance noted for Monocle and TSCAN could be attributed to the lack of controlling for batch-effects.

3. A major limitation of Tempora is that cells are collapsed into populations and the populations are ordered in a trajectory, rather than the cells themselves, thereby losing the single-cell resolution of inferred pseudotemporal ordering. It is unclear why Tempora elects to collapse cells into populations given the dynamic and continuous nature of developmental processes. A discussion of the benefits and limitations of this approach is warranted.

4. While the authors note over 70 available trajectory inference approaches have been developed, only two (Monocle and TSCAN) are directly compared to Tempora. While it is beyond the scope of this paper to compare Tempora to all published trajectory inference methods, an explanation of why these two particular methods were chosen is warranted.

5. Only two time-series datasets were evaluated here (HSMM and murine cortex), though more are available, including Zebrafish (Farrell et al, Science 2018) and human preimplantation embryos (Petropoulos, Cell 2016) to name a few. Notably, the two time-series datasets evaluated by Temporal currently have 271 and 6,000 cells respectively while approximately 40,000 cells are available in the Zebrafish dataset, spanning many more distinct lineages and complex trajectories. Particularly as larger and larger scRNA-seq datasets are generated, it will be important to assess Tempora on a large dataset to better evaluate its scalability, usability, and accuracy for more complex trajectories.

---

Minor comments:

1. The authors note that Tempora begins with an optimized clustering of scRNA-seq data and that over or under clustering may result in inaccurate trajectories. Readers and users would greatly benefit from some general guidelines to help them determine whether their data are over or under clustered. The current level of description in the Methods and Discussion requires further elaboration.

2. The authors note that Tempora assumes that cell differentiation progresses unidirectionally from stem/progenitor states to differentiated states. It seems that application to aberrant disease programs such as cancer where these assumptions are violated would produce erroneous results. A discussion of this caveat should be noted in the discussion to mitigate user error.

3. Without lineage tracing data, it is unclear how the current gold-standard was established. Is it resolution of the gold-standard at the population level or single-cell resolution? Please clarify.

4. The level of detail in terms of parameters used for Harmony, Monocle, TSCAN currently presented in the Methods section needs further elaboration to ensure reproducibility.

Reviewer #2: The manuscript by Tran and Bader proposed a computational pipeline, Tempora, which builds cell trajectories for time-series scRNA-seq data. Comparing with existing pseudo trajectory reconstruction methods for snapshot scRNA-seq data, Tempora takes advantages of both pathway information and the collected experimental temporal information of time-series scRNAseq data to connect and order cell clusters/types across time points. Tempora requires the input of time-series scRNAseq data with the batch effects being removed and the cells being well clustered. Tempora takes the average gene expression profiles, or centroids, of all clusters, and applies the existing method of ARACNE, which uses pathway enrichment information, on cell clusters/types to inference the network. After then, Tempora uses available temporal information from the input data to determine edge directions. The authors evaluated Tempora on two time-series scRNAseq data sets and demonstrated that Tempora can accurately predict the lineages, and illustrated that Tempora outperforms Monocle2 and TSSCAN methods, both of which are designed for snapshot scRNAseq data.

Overall, Tempora combines existing tools and takes extra information than the state-of-the art methods for snapshot scRNAseq data. However, the method is not clear stated and the results are not convincing.

Major points:

(1) There are already a number of computational methods developed for time-series scRNAseq data. For example, TASIC (Bioinformatics 2017: https://doi.org/10.1093/bioinformatics/btx173), SINCERITIES (Bioinformatics 2018: https://doi.org/10.1093/bioinformatics/btx575), CSHMM (Bioinformatics 2019: https://doi.org/10.1093/bioinformatics/btz296), TSEE (BMC Genomics 2019: https://doi.org/10.1186/s12864-019-5477-8),Waddington-OT (Cell 2019: https://doi.org/10.1016/j.cell.2019.01.006). However, the authors did not cite and compare with any of them.

(2) The novelty and strength of Tempora are not clear state. In order to inference cell cluster networks by the existing method ARACNE, Tempora has to treat single-cell data as bulk sequencing data by taking the averaged gene expression profiles, or centroids, of all clusters. A lot of information, such as the cell heterogeneity information (stochasticity of cells) offered by single-cell sequencing are not modeled and lost.

(3) Please clear state why pathway information can help to identify cell cluster/type transition processes. The pathway information can be incomplete for the biological processes studied, will the method still work? The single cell data have a high rate of drop out event. Will the results for the pathway enrichment analysis be sensitive to drop out event?

(4) How many parameters does Tempora have? How sensitive are they to the results? For example, in the implementation of ARACNE, is there any score for removing the edges?

(5) Is Tempora sensitive to clustering results? To compare with Moncle2 and TSCAN, when using the same clustering results by Moncle2 and TSCAN, can Tempora still outperforms them?

Minor points:

(1) Since ARACNE is a key component of Tempora, please provide a detailed description of ARACNE in the supplementary.

(2) It is not clear how the time-dependent pathways are detected? For example, as n Fig 2b,c

Reviewer #3: The authors present a new method to infer trajectories fron single-cell data, that first calculates a pathway enrichment score of clusters, uses the similarity in this enrichment to connect these clusters, and finally determines a direction for the edges between clusters using time series information. The manuscript is well written and results/methods are clearly described. The method is also well implemented as an R package, with good documentation and a nice tutorial. However, there are several flaws regarding the benchmarking and evaluation that, in my opinion, still need to be addressed.

Major comments:

- As the authors discuss, most current TI methods are integrated by the Dynverse group. Why then did the authors only compare their method to of the oldest methods - Monocle and TSCAN - both of which didn't really perform that well in their benchmarking study? (10.1038/s41587-019-0071-9). The benchmarking really needs to be improved on several facets:

- More methods need to be included, which performed the best according to this benchmarking study and/or are the current state-of-the-art (e.g. Monocle 3 and PAGA)

- Other methods that also use temporal information should be evaluated. Waddington-OT is an example of this (there are probably several others in the Dynverse libraries)

- Tempora does not output pseudotime values for each of the cells, so I understand that using the "Cell positions" metric in the Dynverse study is difficult. Nontheless, it would be worthwhile to try it out, given that these authors of this benchmarking study also did apply it on methods that only output clusters

- A comparison on 2 datasets is quite limited, especially given the sheer number of datasets available nowadays. I understand that time series information should be available, but there are also plenty of time series experiments, especially in the cell reprogramming field (e.g. several of the Treutlein lab's papers)

- A benchmark on synthetic data can also be very valuable, I think both PROSSTT and the generator from Dynverse output time series data (or can be adapted in a way that they can).

- How scalable is Tempora with increasing number of cells? I guess this is driven by the speed of the initial steps (pathway enrichment and clustering). How does this compare to other methods? (especially the most scalable ones from aforementioned benchmarking study)

- "Tempora assumes that user input includes an optimized clustering solution for their data." In my experience, the clustering is the step that has the largest influence on the quality of an inferred trajectory. If I understand correctly, the authors used the same clustering strategy for both datasets, i.e. using scClustViz and selecting the # of clustering as described in the methods. Do the authors suggest that the users always use this strategy? How can we be sure that this clustering is not the reason why Tempora performs better than Monocle and TSCAN? Why didn't the authors provide this clustering to either of these methods, or alternatively use the clustering present in either of these methods for Tempora?

- "To calculate the mismatch score between a pair of graphs, we first label each cluster in the inferred trajectory with the cell type(s) it contains, based on expression of a set of wellknown marker genes." Does this mean that this was done separately for the clusters obtained by Tempora, and the clusters obtained by Monocle and TSCAN? This looks like a very arbitrary way to label clusters, and can easily add a bias into this benchmark which is supposed to be objective. Why didn't the authors use a method that can label cells/clusters automatically? (many of them are available, some of which allow you to provide a set of markers for each cell type). This would make the analysis more objective.

- How would the accuracy score of Tempora change if you would determine the direction of the edges in the same way as you did for Monocle and TSCAN (as described in the methods)? The main selling-point of Tempora is that it uses time information, so this is an important experiment to check whether the temporal information adds something (compared to determining the direction using marker genes)

- It seems to me that the "similarity threshold" is an important parameter, that depends on the dataset. What value did the authors choose for the datasets? Was this optimized? If so, how can we be sure that the benchmark is not biased towards Tempora given that the parameters were not optimized for Monocle and TSCAN?

Minor comments:

- "The central concept used by most of these methods to infer temporal cell trajectories is that cells that are close to each other in time (e.g. cells that are closely related to each other via differentiation) will have similar transcriptomes. However, sometimes actual time point information is available to use for this purpose, in which case it should be used directly." This section of the Author Summary is confusing to me. What is meant by "time" here, pseudotime or real-life time? Most trajectory inference methods do not assume that cells are close to each other in real-life time (i.e. snapshot data). I think it should be clearly defined how cells can be related in time, i.e. if their differentiation process happens in a synchronous manner (i.e. time-series data).

- It's a bit weird that Monocle v1 (ICA/MST) is discussed in detail in the introduction, but then Monocle v2 is used in the evaluation. Also, it should always be clearly stated which of these two methods are meant, because the name "Monocle" is used for both v1 and v2 throughout the manuscript (especially given that v3 also exists, which can be even more confusing for people new to the field).

**Have all data underlying the figures and results presented in the manuscript been provided?**

Reviewer #1: Yes

Reviewer #2: Yes

Reviewer #3: Yes

PLOS authors have the option to publish the peer review history of their article (what does this mean?). If published, this will include your full peer review and any attached files.

Reviewer #1: Yes: Jean Fan

Reviewer #2: No

Reviewer #3: No

---

## [Decision Letter · Decision Letter 1]

6 Jul 2020

Dear Ms Tran,

Thank you very much for submitting your manuscript "Tempora: cell trajectory inference using time-series single-cell RNA sequencing data" for consideration at PLOS Computational Biology. As with all papers reviewed by the journal, your manuscript was reviewed by members of the editorial board and by several independent reviewers. The reviewers appreciated the attention to an important topic. Based on the reviews, we are likely to accept this manuscript for publication, providing that you modify the manuscript according to the review recommendations.

Sincerely,

Qing Nie

Associate Editor

PLOS Computational Biology

Thomas Lengauer

Methods Editor

PLOS Computational Biology

[LINK]

Reviewer's Responses to Questions

**Comments to the Authors:**

Reviewer #1: Executive Summary:

In this revision, the authors have expanded on the number of trajectory inference methods compared and also included a larger murine cerebellar dataset for benchmarking. The authors have addressed my previous concerns. I have the following set of minor comments:

1. While the authors have included new analyses highlighting the importance of batch correction prior to trajectory inference, it remains unclear whether the inferior performance noted for Monocle for example (and also the additional methods) could be attributed to inferior batch correction rather than the trajectory inference itself. Presumably, a minimum spanning tree could be constructed on the Harmony-corrected PCs, so a comparison based on the same batch-correction across trajectory inference methods should be achievable.

2. With the new larger murine cerebellar dataset, the authors now note that "when applied on the ~19,000 gene x ~55,000 cell gene mouse cerebellar development expression matrix, Tempora completes in an average of 60 seconds, while Monocle 3 takes 1700 seconds on a modern personal computer." While this is a notable improvement in computational efficiency, does Tempora scale linearly with the number of genes or cells? Or will the runtime growth be exponential with the number of clusters? It may be useful for readers as the size and complexity of single-cell RNA-seq datasets increases.

Reviewer #2: The authors have done an excellent job in revision of the manucript and all my concerns have been addressed. I have one comment for the authors. The dimentionality reduction is very important for the downstream analysis of time series scRNA-seq data, but few study has realized it. I hope the author can take a look at the TSEE algorithm (BMC Genomics 2019: https://doi.org/10.1186/s12864-019-5477-8), the first dimentionality reduction algorithm that can incooperate the time stage information of time series scRNA-seq data. It outperforms existing methods (e.g. PCA, t-SNE) in preserving local and global structures as well as enhancing the temporal resolution of samples. It can uncover the subtle gene expression patterns, facilitating further downstream analysis.

Reviewer #3: The authors have fulfilled all my concerns, and by doing that, the authors have improved the methodology and scientific soundness of their study. I have no further concerns.

**Have all data underlying the figures and results presented in the manuscript been provided?**

Reviewer #1: Yes

Reviewer #2: Yes

Reviewer #3: Yes

PLOS authors have the option to publish the peer review history of their article (what does this mean?). If published, this will include your full peer review and any attached files.

Reviewer #1: **Yes: **Jean Fan

Reviewer #2: **Yes: **Lin Wan

Reviewer #3: No
---

## [Editor Report · Decision Letter 2]

29 Jul 2020

Dear Ms Tran,

We are pleased to inform you that your manuscript 'Tempora: cell trajectory inference using time-series single-cell RNA sequencing data' has been provisionally accepted for publication in PLOS Computational Biology.

Best regards,

Qing Nie

Associate Editor

PLOS Computational Biology

Thomas Lengauer

Methods Editor

PLOS Computational Biology

---

## [Editor Report · Acceptance letter]

4 Sep 2020

PCOMPBIOL-D-19-02083R2 

Tempora: cell trajectory inference using time-series single-cell RNA sequencing data

Dear Dr Bader,

I am pleased to inform you that your manuscript has been formally accepted for publication in PLOS Computational Biology. Your manuscript is now with our production department and you will be notified of the publication date in due course.

With kind regards,

Laura Mallard
